# Mouse gingival single-cell transcriptomic atlas identified a novel fibroblast subpopulation activated to guide oral barrier immunity in periodontitis

Takeru Kondo[1,2], Annie Gleason[1,3], Hiroko Okawa[1,2], Akishige Hokugo[1,4], Ichiro Nishimura[1]*

[1]Weintraub Center for Reconstructive Biotechnology, UCLA School of Dentistry, Los Angeles, United States; [2]Division of Molecular and Regenerative Prosthodontics, Tohoku University Graduate School of Dentistry, Sendai, Japan; [3]UCLA Bruin in Genomics Summer Program, Los Angeles, United States; [4]Regenerative Bioengineering and Repair Laboratory, Division of Plastic and Reconstructive Surgery, Department of Surgery, David Geffen School of Medicine at UCLA, Los Angeles, United States

*For correspondence:
inishimura@dentistry.ucla.edu

**Abstract** Periodontitis, one of the most common non-communicable diseases, is characterized by chronic oral inflammation and uncontrolled tooth supporting alveolar bone resorption. Its underlying mechanism to initiate aberrant oral barrier immunity has yet to be delineated. Here, we report a unique fibroblast subpopulation activated to guide oral inflammation (AG fibroblasts) identified in a single-cell RNA sequencing gingival cell atlas constructed from the mouse periodontitis models. AG fibroblasts localized beneath the gingival epithelium and in the cervical periodontal ligament responded to the ligature placement and to the discrete topical application of Toll-like receptor stimulants to mouse maxillary tissue. The upregulated chemokines and ligands of AG fibroblasts linked to the putative receptors of neutrophils in the early stages of periodontitis. In the established chronic inflammation, neutrophils, together with AG fibroblasts, appeared to induce type 3 innate lymphoid cells (ILC3s) that were the primary source of interleukin-17 cytokines. The comparative analysis of *Rag2-/-* and *Rag2-/-Il2rg-/-* mice suggested that ILC3 contributed to the cervical alveolar bone resorption interfacing the gingival inflammation. We propose the AG fibroblast–neutrophil–ILC3 axis as a previously unrecognized mechanism which could be involved in the complex interplay between oral barrier immune cells contributing to pathological inflammation in periodontitis.

## eLife assessment

The findings of this article provide **valuable** information on the changes of cell clusters induced by chronic periodontitis. The observation of a new fibroblast subpopulation, named AG fibroblasts, is interesting, and the strength of evidence presented is **solid**.

## Introduction

The oral mucosa is one of the most active barrier tissues in the human body (*Abusleme et al., 2013*; *Belkaid and Harrison, 2017*), and oral barrier immunity acts as a crucial surveillance system to achieve the homeostasis (*Moutsopoulos and Moutsopoulos, 2018*). However, once the gingival inflammation progresses, connective tissues supporting the cervical area of the dentition are subjected to a

localized and severe degeneration, resulting in tooth loss and disruption of the maxillofacial structure (*Lamont et al., 2018*). Periodontitis is not only the most frequent cause of tooth loss in adults (*Helal et al., 2019*), but also, globally, ranks among the most significant contributors to poor health and decreased quality of life, imposing substantial economic and healthcare burdens (*Peres et al., 2019*).

The postulated pathological framework of progressive gingival inflammation has been reconstructed from animal models (*Lin et al., 2021*; *Kondo et al., 2022*), harvested immune cell analyses, and oral microbial associations (*Lamont et al., 2018*; *Hajishengallis and Chavakis, 2021*; *Shokeen et al., 2022*). For example, the ligature-induced periodontitis mouse model has shown the abundant recruitment and excessive activation of neutrophils (*Hajishengallis, 2020*) and pathological induction of interleukin (IL)-17-secreting proinflammatory effector CD4$^+$ T helper (Th)17 cells (*Hasiakos et al., 2021*). It has been hypothesized that these pathological immune cells activate osteoclasts, leading to induction of periodontal alveolar bone resorption (*Cekici et al., 2014*; *Sokol and Luster, 2015*).

Prior studies have shown that chemokines are involved in the trafficking and activation of inflammatory cells during both homeostasis and disease-associated inflammation (*Sokol and Luster, 2015*). Chemokines containing disulfide cysteine–cysteine (CC) and cysteine–X–cysteine (CXC) molecular signatures act as ligands for cellular receptors that modulate inflammatory signaling pathways (*Hughes and Nibbs, 2018*). Such chemokine–receptor networks developed among immune cells direct the migration of inflammatory cells, potentially amplifying the resulting tissue damage. However, increasing evidence suggests that in addition to immune cells, barrier tissue stromal cells are involved in innate immune cell regulation (*Hodzic et al., 2017*; *Ina et al., 2005*; *Pinchuk et al., 2008*; *Cheng et al., 2018*). It has been reported that oral fibroblasts secrete cytokines and chemokines in response to microbial stimuli or to a proinflammatory environment (*Kondo et al., 2022*; *Williams et al., 2021*). Therefore, a comprehensive elucidation of chemokine–receptor signaling networks will likely need to include all types of gingival cells.

In this study, to better understand the pathways contributing to chronic oral inflammation, we constructed the gingival single-cell transcriptomic atlas of the mouse periodontitis models. Here we report a novel subpopulation of fibroblasts activated to guide chronic inflammation (AG fibroblasts). Our findings suggest that chemokines and ligands derived from AG fibroblasts could bind to and activate receptors involved in neutrophil and lymphocytes including *Cd4*$^-$ innate lymphoid cells (ILCs) predominantly producing proinflammatory IL-17 cytokines. This study identified the AG fibroblast–neutrophil–ILC3 axis as a previously unrecognized mechanism which could be involved in the complex interplay between oral barrier immune cells contributing to pathological inflammation in periodontitis.

## Results

### Alterations in major cell-type proportions during periodontitis development

The alveolar bone and gingival tissue surrounding the maxillary second molar of C57BL/6J mice presented healthy connective tissue supporting the dentition with minimal CD45+ immune cell infiltration and well-developed collagen architecture (*Figure 1A*, *Figure 1—figure supplement 1A*). The placement of a ligature around the maxillary left second molar (*Abe and Hajishengallis, 2013*) induced a localized and small gingival connective tissue degradation and infiltration of CD45+ immune cells near the cervical area (*Figure 1B*). The noticeable gingival defect and more prominent though still localized CD45+ immune cell infiltration developed after day 3 following ligature placement (*Figure 1C*, *Figure 1—figure supplement 1A and B*). On day 7, the gingival connective tissue was largely degraded, and chronic inflammation was developed (*Figure 1D*, *Figure 1—figure supplement 1A and B*). Three-dimensional (3D) reconstruction of micro-computed tomography (microCT) images (*Figure 1—figure supplement 1C*) further revealed a reduction in alveolar bone height starting from day 3, which progressively increased on day 7 (*Figure 1—figure supplement 1D and E*). Overall, the observed pattern of periodontal tissue degradation was consistent with that reported in previous studies (*Tamura et al., 2021*; *Marchesan et al., 2018*).

Left-side palatal gingiva tissue was harvested from mice on day 0 (i.e., healthy gingiva without ligature placement) and on days 1, 4, and 7 after ligature placement, and gingival cells were dissociated for single-cell RNA sequencing (scRNA-seq) (*Kondo et al., 2022*; *Okawa et al., 2022a*). On days 0 and 1, the major cell types identified included epithelial cells expressing cadherin 1 (*Cdh1*) (*Groeger and*

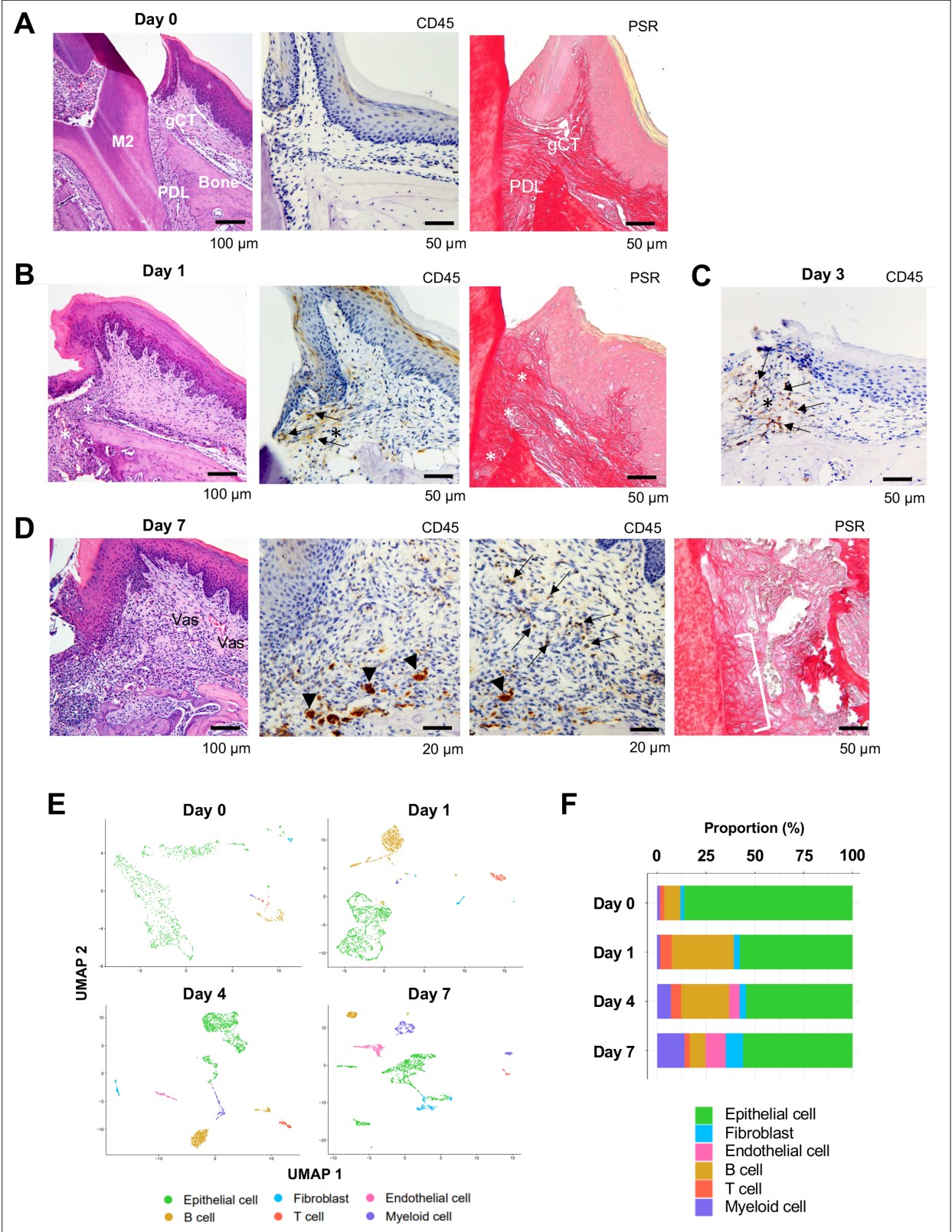

**Figure 1.** Changes in proportions of major cell types during ligature-induced periodontitis development in mice. (**A**) Mouse maxillary second molar (M2) has multi-roots and supported by alveolar bone (Bone), periodontal ligament (PDL), and gingiva connective tissue (gCT). The oral barrier immunity is not constitutively activated as evidenced by the lack of CD45+ immune cells. Picrosirius red (PSR)-stained collagen fibers connected the root surface and alveolar bone in the PDL and organized as dense parallel bundles in gCT. (**B**) One day (day 1) after a ligature (5.0 silk suture) was placed around

*Figure 1 continued on next page*

*Figure 1 continued*

M2, the cervical PDL and gCT demonstrated a localized connective tissue degradation (*), where CD45+ immune cells clustered (arrows). PSR-stained collagen architecture immediately under the ligature lost the thick collagen bundle structure (*). (**C**) Day 3 of ligature placement exhibited localized but increased CD45+ immune cell clustering adjacent to the collagen degradation area (*). (**D**) Day 7 of ligature placement, PDL, and gCT tissue degradation increased with inflammatory vascularization (Vas). CD45+ myeloid cells (arrowheads) were observed near the alveolar bone surface and CD45+ lymphocytes (arrows) infiltrated the gCT area. PSR staining lost the typical collagen pattern in gCT and PDL, and a remnant of degraded PDL collagen fiber (white bracket) was attached to the tooth surface. (**E**) Single-cell RNA sequencing (scRNA-seq) *t*-distributed stochastic neighbor embedding (*t*-SNE) projection plots showing the major cell types present within gingival tissue during periodontitis development on days 0, 1, 4, and 7. Colors indicate cell type as follows: green, epithelial cells; blue, fibroblasts; pink, endothelial cells; yellow, B cells; red, T cells; and purple, myeloid cells. (**F**) Proportion plots showing the relative amounts of each major cell type on days 0, 1, 4, and 7.

The online version of this article includes the following figure supplement(s) for figure 1:

**Figure supplement 1.** Gingival defect formation and alveolar bone loss in the ligature-induced periodontitis model in mice.

**Figure supplement 2.** Identification of major cell types in mouse gingival tissue during periodontitis development by single-cell RNA sequencing (scRNA-seq).

*Meyle, 2019*) and type XVII collagen (*Col17a1*) (*Kamaguchi et al., 2018*), fibroblasts expressing type I collagen (*Col1a1*) (*Takahashi et al., 2019*) and lumican (*Lum*) (*Lallier et al., 2005*), B cells expressing membrane spanning 4 domains A1 (*Ms4a1*) (*Bruno et al., 2010*) and cluster of differentiation 79A (*Cd79a*) (*Sparger et al., 2018*), T cells expressing epsilon subunit of T cell receptor complex (*Cd3e*) (*Alcover et al., 2018*) and cluster of differentiation 5 (*Cd5*) (*Fujihashi et al., 1989*), and myeloid cells expressing lysozyme 2 (*Lyz2*) (*Cross et al., 1988*) and integrin subunit alpha M (*Itgam*) (*Okubo et al., 2016*; *Figure 1E* and *Figure 1—figure supplement 2*). On days 4 and 7, an additional endothelial cell fraction expressing selectin P (*Selp*) (*Gotsch et al., 1994*) and selectin E (*Sele*) (*Komatsu et al., 2012*) emerged (*Figure 1E* and *Figure 1—figure supplement 2*), suggesting increased inflammatory neovascularization with the progression of periodontal inflammation. The proportion of B cells was increased on day 1, and the proportion of myeloid cells increased progressively from days 4–7 (*Figure 1F*). In addition, the proportion of fibroblasts was increased on day 7 (*Figure 1F*).

## Fibroblasts activated to guide leukocyte migration in periodontitis development

We previously identified two distinct subpopulations of gingival fibroblasts, differentiated by expression of type XIV collagen (*Col14a1*) (*Kondo et al., 2022*), and these were also detected in our current scRNA-seq data from days 0–7 (*Figure 2A*). Gene Ontology (GO) enrichment analysis of the *Col14a1*-expressing fibroblast subpopulation revealed expression of major gene clusters related to immune regulation, including 'Regulation of leukocyte migration' (*Figure 2B*). This immune regulatory phenotype appears to be unique to *Col14a1*-expressing fibroblasts, which are therefore referred to as 'fibroblasts activated to guide leukocyte migration' or AG fibroblasts. The other subpopulation of *Col1a1*-expressing fibroblasts appeared to keep typical fibroblastic features and is thus referred to as KT fibroblasts (*Figure 2C*). GO analysis suggested that KT fibroblasts primarily maintained connective tissue remodeling but did not express an immune regulatory phenotype. An additional fibroblast subpopulation expressing smooth muscle actin alpha 2 (*Acta2*) was detected on day 7, and these were identified as myofibroblasts (MF) (*Sasaki et al., 2020*; *Figure 2A*). We found that the proportions of KT and AG fibroblasts were equal on day 0 and on days 1 and 4 after ligature placement (*Figure 2E and F*). However, on day 7, the proportion of AG fibroblasts decreased, and the MF fraction emerged (*Figure 2E and F*).

## AG fibroblasts and immune surveillance in early periodontitis development

To further characterize AG fibroblasts and their 'Regulation of leukocyte migration' phenotype, we analyzed expression of CC motif chemokine ligands (CCLs) and CXC motif chemokine ligands (CXCLs) within fibroblast subpopulations in our scRNA-seq dataset (*Cekici et al., 2014*). Results show that AG fibroblasts activated expression of *Ccl8, Ccl11, Ccl19, Cxcl1, Cxcl10,* and *Cxcl12* immediately after ligature placement on days 1 and 4 (*Figure 3A*). Chemokine expression levels then decreased on day 7, suggesting that AG fibroblasts might initiate early leukocyte migration into gingival tissue. Given that activation of Toll-like receptors (TLR) is known to increase chemokine expression, we further

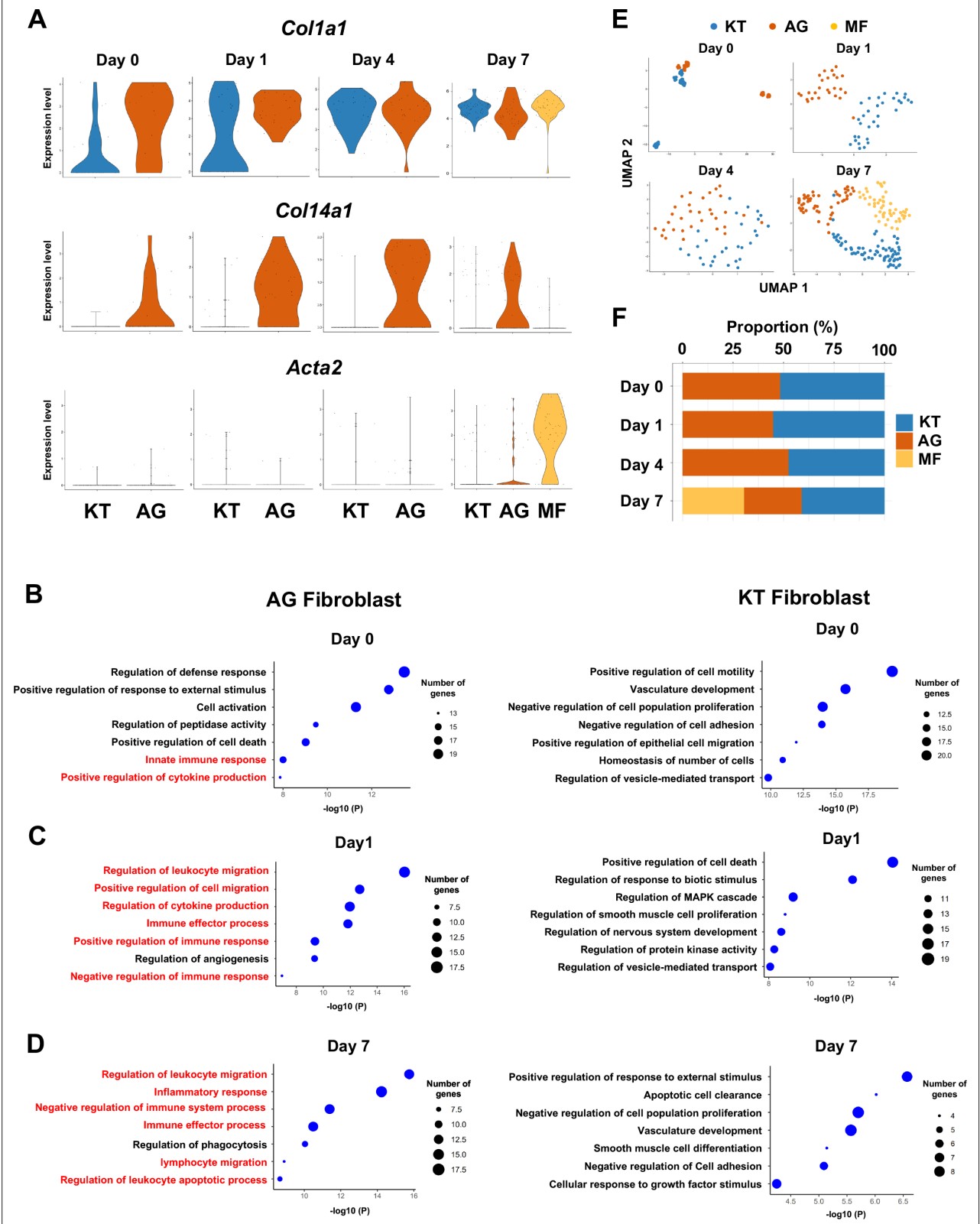

**Figure 2.** Fibroblasts activated to guide leukocyte migration (AG fibroblasts) are one of three fibroblast subpopulations in gingival tissue during periodontitis development. (**A**) Violin plots showing gene expression levels of type I collagen (*Col1a1*), type XIV collagen (*Col14a1*), and smooth muscle aortic actin 2 (*Acta2*) in gingival fibroblast subpopulations during periodontitis development. AG, AG fibroblasts; KT, 'keeping typical phenotype' fibroblasts; MF, myofibroblasts. Gene Ontology (GO) enrichment analysis of the biological functions of AG fibroblasts and KT fibroblasts on day 0

*Figure 2 continued on next page*

*Figure 2 continued*

without ligature placement (**B**) and on day 1 (**C**) and day 7 (**D**) after ligature placement. Gene clusters related to immune regulation (red) were identified in AG fibroblasts, and these clusters dominate after ligature placement. (**E**) *t*-distributed stochastic neighbor embedding (*t*-SNE) projection plots showing fibroblast subpopulations in gingival tissue during periodontitis development. Colors indicate cell type as follows: blue, KT fibroblasts; red, AG fibroblasts; and yellow, MFs. (**F**) Proportion plots showing the relative amounts of each fibroblast subpopulation on days 0, 1, 4, and 7.

assessed expression of TLRs in fibroblast subpopulations. We found that AG fibroblasts displayed temporal activation of *Tlr2*, *Tlr3*, and *Tlr4* expression on days 1 and 4 (*Figure 3B*). Similarly, the TLR downstream genes *Myd88*, *Irak1*, *Map3k7*, and *RelA* were also expressed on days 1 and 4 (*Figure 3B*). These data suggest that AG fibroblasts may guide the establishment of an early inflammatory environment within the gingiva and thereby promote periodontitis pathogenesis.

To validate the presence of AG fibroblasts, day 1 gingiva and periodontal tissue histological sections were subjected to immunohistochemistry with antibodies against COL14A1 and CXCL12 (*Figure 3C*). We detected COL14A1- and CXCL12-positive AG fibroblasts localized near gingival epithelial cells in the connective tissue papillae and free gingiva, as well as in the cervical zone of the periodontal ligament (PDL) space (*Figure 3C*). We note that this localization pattern of AG fibroblasts appears to be highly suitable for early immune surveillance during periodontitis pathogenesis. Thus, we have hypothesized that AG fibroblasts initially sense the pathological stress including oral microbial stimuli through TLRs and secrete inflammatory signals through chemokine expression.

## AG fibroblasts induced by maxillary topical application (MTA) of unmethylated cytidine phosphate guanosine oligonucleotide (CpG ODN) and of *Porphyromonas gingivalis* lipopolysaccharide (LPS)

We previously reported that the microbial composition of the mouse ligature did not mirror the human oral microbial composition (*Kondo et al., 2022*). To mitigate this critical discrepancy in the mouse periodontitis model, we developed the MTA model. The MTA model applied oral microbial biofilm directly to the maxillary gingiva and held under a custom-made oral appliance (*Figure 4A*) for 1 hr. We previously reported that human oral microbial biofilm, but not planktonic microbes, induced initial gingival tissue degradation in the MTA model, suggesting that extracellular components of human oral biofilm could play an important role in the initiation of periodontitis (*Kondo et al., 2022*). In this study, we used the MTA model without the placement of a ligature to further characterize the behavior of AG fibroblasts. This study discretely applied ligands of TLR9 and TLR2/4: unmethylated CpG ODN and *P. gingivalis* LPS, respectively. Four days after the topical application, mouse maxillary tissue was harvested for histological analysis and for scRNA-seq (*Figure 4B*; *Kondo et al., 2022*). CpG ODN induced the expression of cathepsin K (Ctsk) in the PDL and gingival connective tissue (*Figure 4C*) validating the previous report (*Kondo et al., 2022*). Furthermore, there were signs of localized Ctsk+ osteoclastic activities on the surface of alveolar bone (*Figure 4C*).

The scRNA-seq gingival cell composition of the MTA of CpG ODN and LPS indicated the early periodontitis pattern (*Figure 4E*) consistent with day 1 scRNA-seq of the ligature-induced periodontitis (*Figure 1F*). The MTA of CpG ODN and LPS identified AG fibroblasts along with KT fibroblasts, whereas the MTA of LPS induced the additional MF (*Figure 4E*). Further analysis demonstrated the expression of CCL and CXCL chemokines by AG fibroblasts activated by the MTA of CpG ODN and LPS (*Figure 4F*). The scRNA-seq of MTA of CpG ODN did not capture the transcriptional activation of *Tlr9*; however, the downstream effector genes associated with TLR9 such as *Myd99*, *Irak1*, and *RelA* were found to be upregulated in AG fibroblasts (*Figure 4G*). The MTA of LPS appeared to upregulate *Tlr2* and *Tlr4* gene transcription of AG fibroblasts and of MF albeit at lesser degrees (*Figure 4G*). These results indicated that AG fibroblasts were activated by CpG ODN and LPS through the MTA model. As such, the extracellular substances of human oral biofilm such as microbial extracellular DNA and LPS might be an important trigger of gingival inflammation.

## Myeloid cell composition and activity during periodontitis development

The infiltration of proinflammatory neutrophils into the gingiva has been extensively characterized in the mouse model of ligature-induced periodontitis (*Hajishengallis, 2020*; *Silva et al., 2019*). Here, we found that macrophages expressing cluster of differentiation 86 (*Cd86*) (*Lam et al., 2014*) and integrin subunit alpha X (*Itgax*) (*Agarbati et al., 2021*) were predominant on day 0 in the myeloid cell fraction

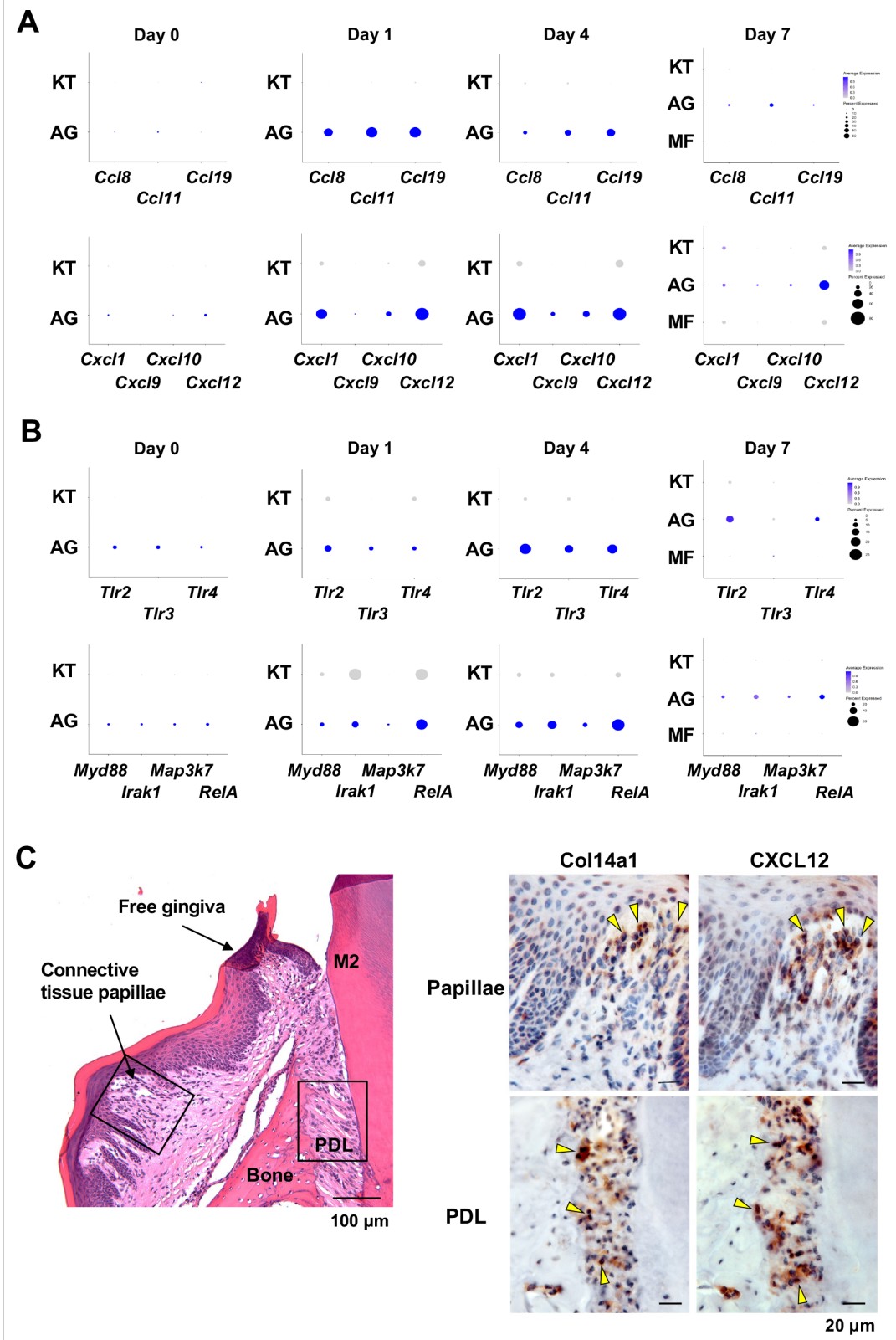

**Figure 3.** AG fibroblasts and immune surveillance in periodontitis development. (**A**) Dot plots depicting expression levels of the CCL genes *Ccl8*, *Ccl11*, *Ccl19*, *Cxcl1*, *Cxcl9*, *Cxcl10*, and *Cxcl12* in gingival fibroblast subpopulations during periodontitis development. (**B**) Dot plots depicting expression levels of the Toll-like receptor (TLR) and related genes *Tlr2*, *Tlr3*, *Tlr4*, *Myd88*, *Irak1*, *Map3k7*, and *Rela* in gingival fibroblast

*Figure 3 continued on next page*

*Figure 3 continued*

subpopulations during periodontitis development. Upregulation of chemokines and TLR-related molecules is predominantly observed in the AG fibroblast subpopulation. (**C**) Hematoxylin and eosin (HE) staining and immunohistochemical (IHC) staining for COL14A1 and CXCL12 in periodontal tissue on day 1; scale bars, 100 μm (HE) and 20 μm, (IHC). Yellow arrows indicate COL14A1- and CXCL12-positive cells in the connective tissue papillae and periodontal ligament (PDL).

from healthy gingiva (*Figure 5A and B*, *Figure 5—figure supplement 1*). In contrast, the proportion of neutrophils expressing CXC motif chemokine receptor (*Cxcr2*) (*Hashim et al., 2021*) and G0/G1 switch gene 2 (*G0s2*) (*Zhang et al., 2017*) increased after ligature placement and during periodontitis development from days 1–7 (*Figure 5A and B*, *Figure 5—figure supplement 1*), suggesting continuous neutrophil infiltration into the early and established gingival lesion. Moreover, after ligature placement, gingival neutrophils upregulated the expression of triggering receptor expressed on myeloid cells 1 (*Trem1*), indicating that these cells are activated and participating in the amplification of inflammatory signals (*Bouchon et al., 2000*; *Dopheide et al., 2013*; *Figure 5C*). Strikingly, the *Trem1*-expressing activated neutrophils also show upregulation of matrix metalloproteinase 9 (*Mmp9*; *Figure 5C*)—a protein associated with extracellular matrix degeneration within gingival tissue (*Corotti et al., 2009*). We further detected expression of tumor necrosis factor (*Tnf*) and transforming growth factor beta 1 (*Tgfb1*) in both macrophages and neutrophils on day 0 and after ligature placement (*Figure 5D*). Collectively, these myeloid cell behaviors are consistent with those reported in prior studies on periodontitis development, thus validating our scRNA-seq data.

Myeloid cells are also known to stimulate other immune cells through the expression of CCL and CXCL chemokines, many of which are associated with periodontitis development (*Souto et al., 2014*). Our scRNA-seq analysis revealed that macrophages expressed *Ccl2*, *Ccl9*, Cxcl4, and *Cxcl6*, and neutrophils expressed *Ccl3*, *Ccl4*, *Ccl6*, *Cxcl2*, and *Cxcl3* throughout periodontitis development (*Figure 5E and F*). These data suggest that chemokines and cytokines produced by macrophages and neutrophils in inflamed tissue may amplify and polarize the immune response toward chronic gingival inflammation.

## Role of AG fibroblasts in myeloid cell activation

The interaction between chemokine ligands and their receptors has been extensively studied. Here, to evaluate the interaction between chemokine ligands strongly expressed by AG fibroblasts and chemokine receptors in innate immune cells, we matched AG fibroblast-expressed chemokines to expression of their putative receptors in myeloid cells. Results show that AG fibroblasts may regulate macrophages via the expression *Ccl8* and *Ccl11*, which encode chemokines that can interact with CC chemokine receptors (CCRs) encoded by *Ccr2* and *Ccr5* in macrophages. Similarly, gene expression signatures suggest that AG fibroblast-mediated neutrophil regulation may occur through CCL8–CCR1, CXCL1–CXCR2, and CXCL12–CXCR4 interactions (*Figure 6A*). Notably, all chemokine–receptor pairs are expressed throughout periodontitis development, although expression of factors mediating the interaction between AG fibroblasts and macrophages was decreased on day 7.

Intercellular communication may occur via other ligand–receptor interactions, which induce downstream target gene expression in recipient cells. Here, we performed NicheNet analysis to identify potential interactions between gingival fibroblasts and myeloid cells, revealing a trend toward increasing ligand–target interactions between fibroblasts and myeloid cells from day 1 to day 7 (*Figure 6B*). On day 1, ligand expression was present primarily in AG fibroblasts, whereas on day 7, MFs also exhibited prominent ligand expression (*Figure 6C*). Analysis of target gene expression revealed that both macrophages and neutrophils expressed receptors capable of interacting with AG fibroblast ligands on day 1. In contrast, on day 7, receptor expression was almost exclusively present in neutrophils, suggesting that these cells are primarily targeted by MFs and AG fibroblasts at later stages of periodontitis development (*Figure 6D*).

The chemokine–receptor interaction was also suggested between AG fibroblasts and myeloid cells in the MTA of CpG DON (*Figure 6E*). However, in the MTA of LPS, macrophage lacked the detectable expression of chemokine receptor and the AG fibroblastic chemokine interaction appeared to be limited to neutrophils (*Figure 6E*).

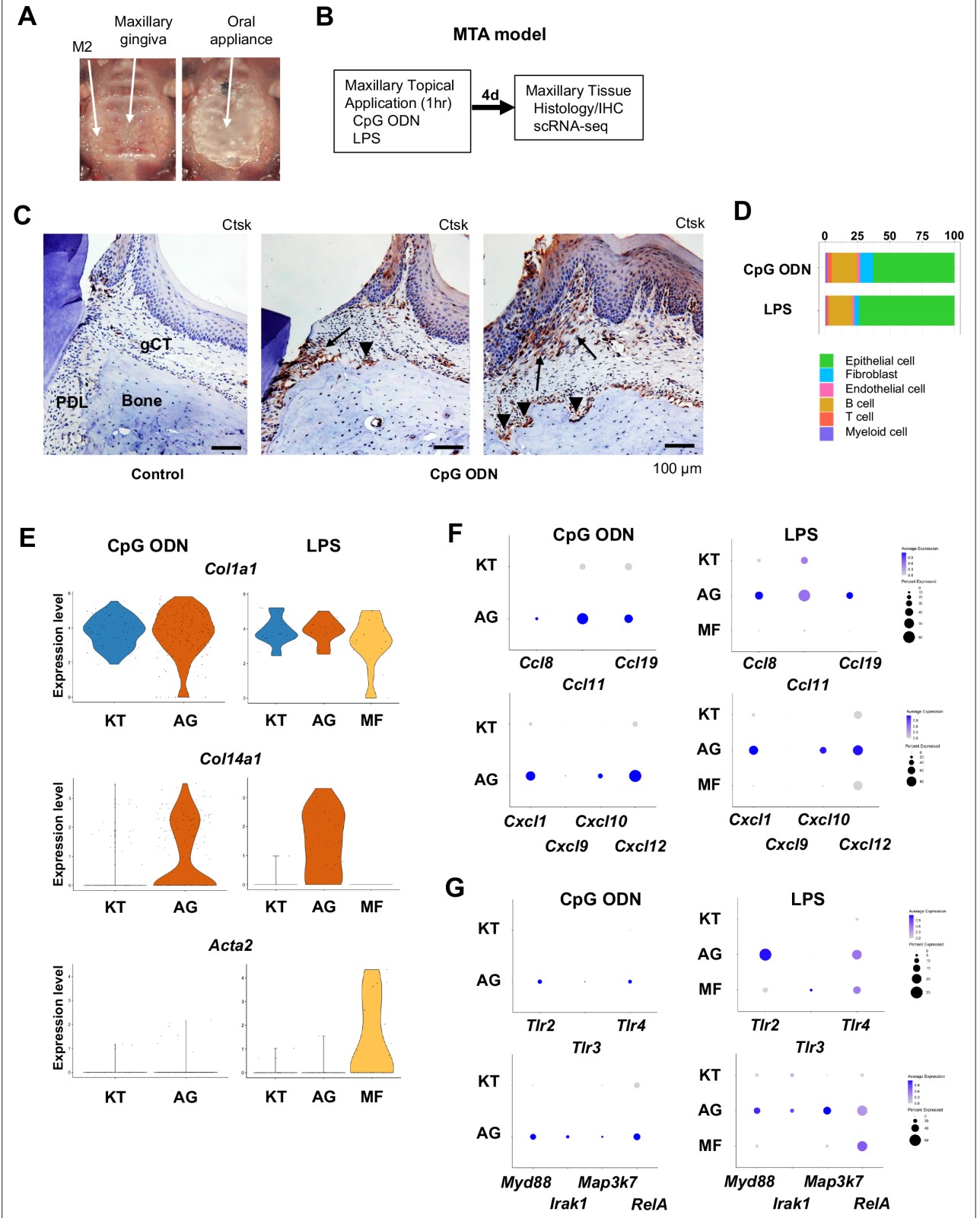

**Figure 4.** Characterization of AG fibroblasts in the discrete maxillary topical application (MTA) model using cytidine phosphate guanosine oligonucleotide (CpG ODN) (TLR9 ligand) and LPS (TLR2/4 ligand). (**A**) The MTA model was developed to discretely test the selected oral microbial pathogens. The selected pathogen was topically applied directly on the maxillary gingiva between the molars and covered by a custom fabricated oral appliance for 1 hr. (**B**) In this study, microbial DNA TLR9 ligand, unmethylated CpG ODN, and TLR2/4 ligand *P. gingivalis* lipopolysaccharide (LPS)

*Figure 4 continued on next page*

*Figure 4 continued*

were selected in the MTA model. After 1 hr exposure, mice were returned to the vivarium for 4 d and the maxillary tissue was harvested for histology or scRNA-seq. (**C**) The MTA of CpG ODN developed localized periodontal ligament (PDL) and gingiva connective tissue (gCT) degradation evidenced by the expression of cathepsin K (Ctsk; arrows). There were signs of localized bone resorption by Ctsk+ osteoclasts (arrowheads) on the surface of alveolar bone (Bone). (**D**) Gingival cell composition by scRNA-seq of the MTA of CpG ODN or LPS revealed early stage of gingival inflammation, equivalent to day 1 of the ligature-induced periodontitis model. (**E**) The fibroblastic gene expression signature revealed the presence of KT and AG fibroblasts by the MTA of CpG ODN, whereas the MTA of LPS induced myofibroblast (MF). (**F**) AG fibroblasts of the MTA of CpG ODN and LPS were activated to express CCL and CXCL chemokines. (**G**) The MTA of CpG ODN did not upregulate *Tlr9*, whereas the MTA of LPS increased the expression of *Tlr2/4* in AG fibroblasts. However, the both the MTA of CpG ODN and LPS increased the expression of TLR downstream molecules.

## Expression of osteoclastogenic cytokines

Macrophage-colony stimulating factor (M-CSF), encoded by *Csf1*, and receptor activator of nuclear factor kappa-Β ligand (RANKL), encoded by *Tnfsf11*, are known to play critical roles in osteoclast induction (*Teitelbaum, 2000*). We found that in the ligature-induced periodontitis model, the expression of M-CSF was detected in various cell types including fibroblasts (*Figure 7A*), whereas the RNAKL expression was more restricted to fibroblasts and T cells (*Figure 7B*). AG fibroblasts consistently expressed M-CSF from day 0 to day 7 and neutrophils started to express M-CSF on days 4 and 7 (*Figure 7C*). The expression of RNAKL was detected in AG fibroblasts but not in myeloid cells (*Figure 7D*). T cells and type 3 innate lymphoid cells (ILC3; see below) were also shown to express M-CSF and RNAKL (*Figure 7C and D*).

The MTA of CpG ODN and LPS suggested the AG fibroblasts were one of the predominant cellular sources of M-CSF (*Figure 7E*). By contrast, the expression of RNAKL was only detected in AG fibroblasts of the MTA of CpG ODN and LPS did not seem to induce RANKL expression in the MTA model (*Figure 7F*). The osteoclastic activity found in the MTA of CpG ODN (*Figure 4D*) might be induced by M-CSF and RANKL derived from AG fibroblasts.

## Helper T (Th) cells, cytotoxic T (Tc) cells, regulatory T (Treg) cells, and innate lymphoid cells (ILCs) in periodontitis development

Our scRNA-seq dataset found that the day 0 healthy gingiva exclusively contained Th cells expressing *Cd4* (*Mahanonda et al., 2018*). However, after ligature placement, Tc cells expressing *Cd8* (*Mahanonda et al., 2018*) emerged, and on day 7, *Cd4⁻Zbtb16⁺* ILCs and *Cd4⁺Foxp3⁺* Treg cells (*Wei et al., 2021*) were also detected (*Figure 8A and B*). The ILCs expressed *Nfil3*, a basic leucine zipper (bZIP) transcription factor required for ILC development (*Eberl et al., 2015*; *Figure 8C*). However, to our surprise, expression of *Rorc*, *Il17a*, and *Il17f* was detected in ILCs (*Figure 8C*), indicating that this gingival subpopulation is predominantly composed of type 3 ILCs (ILC3s). ILC3s and Th17 cells share similar regulatory functions. However, the role of ILC3s in the development of periodontitis has not been fully deciphered.

## ILC3s are critical for cervical alveolar bone resorption in the mouse periodontitis model

We thus examined the role of ILC3s in periodontitis pathogenesis by measuring ligature-induced gingival defects and alveolar bone resorption in *Rag2⁻/⁻* mice, which lack functional B, Th, and Tc cells, and *Rag2*–IL-2 receptor common gamma (*Il2rg*) double-knockout mice, lacking all lymphocytes including ILCs. After ligature placement, we found that alveolar bone loss was decreased in *Rag2⁻/⁻* mice and nearly eliminated in *Rag2⁻/⁻Il2rg⁻/⁻* mice (*Figure 8D*). MicroCT image analysis indicated a better perseveration of alveolar bone structure in *Rag2⁻/⁻Il2rg⁻/⁻* mice, relative to the other groups (*Figure 8E*, *Figure 8—figure supplement 1A and B*). However, gingival defects developed similarly in wild-type (WT), *Rag2⁻/⁻*, and *Rag2⁻/⁻Il2rg⁻/⁻* mice (*Figure 8—figure supplement 1C and D*).

Histologically, osteoclastic resorption lacunae were observed on the alveolar bone surface at the cervical PDL and tooth apex PDL zones in WT mice (*Figure 8F and G*). In addition, we detected a significant decrease in the number of tartrate-resistant acid phosphatase (TRAP)-positive osteoclasts in the cervical PDL zone of *Rag2⁻/⁻Il2rg⁻/⁻* mice (*Figure 8H*) and in the apical PDL zone of both *Rag2⁻/⁻* and *Rag2⁻/⁻Il2rg⁻/⁻* mice (*Figure 8I*). Collectively, these data suggest that ILC3s, not Th17 cells, are responsible for cervical alveolar bone resorption in the mouse periodontitis model, a pathological phenotype consistent with human disease (*Figure 8—figure supplement 2A and B*).

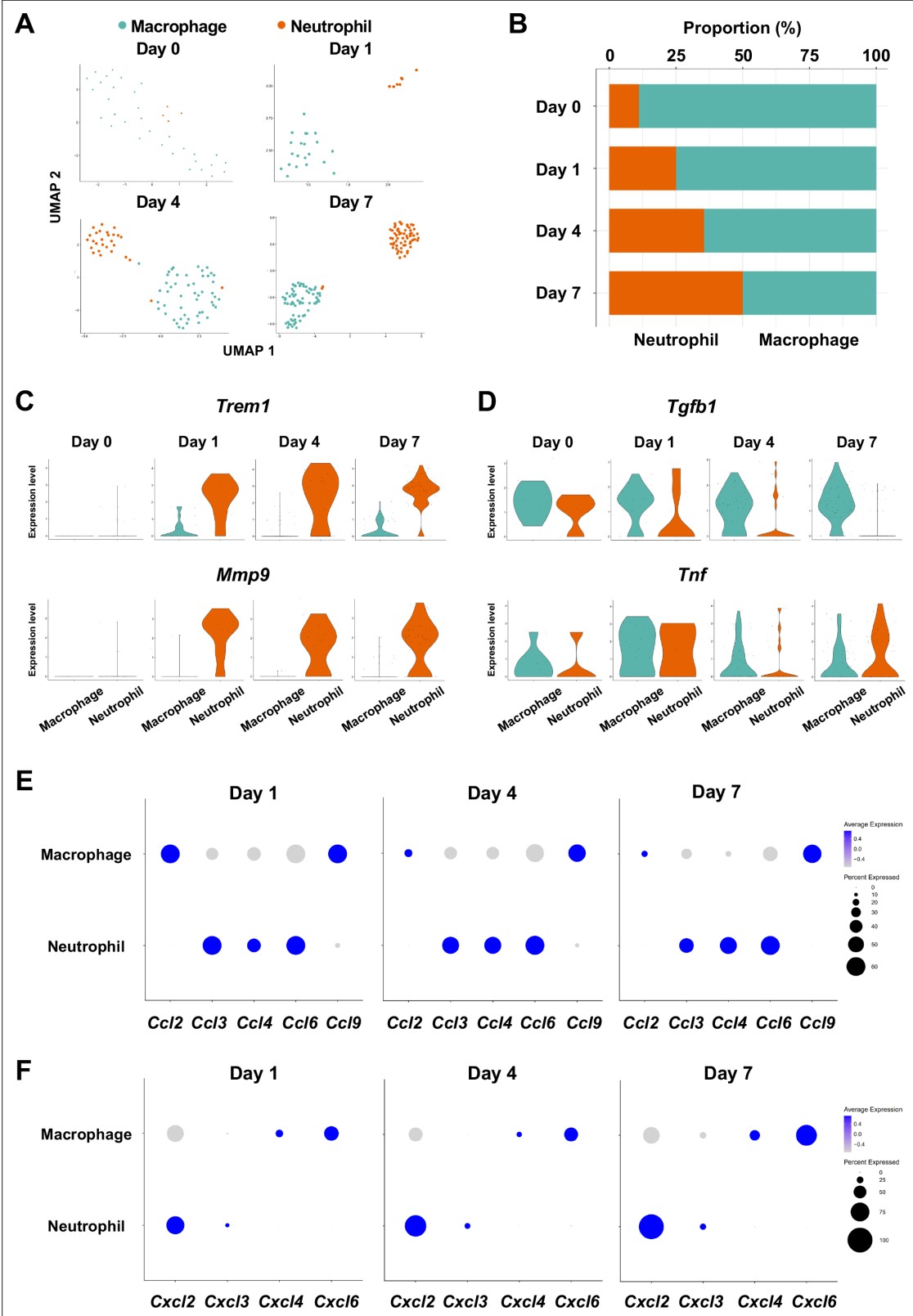

**Figure 5.** Myeloid cell composition and activity in gingival tissue during in periodontitis development. (**A**) *t*-distributed stochastic neighbor embedding (*t*-SNE) projection plots showing myeloid cell subpopulations in gingival tissue during periodontitis development on days 0, 1, 4, and 7. Colors indicate cell type as follows: green, macrophages; and red, neutrophils. (**B**) Proportion plots showing the relative amounts of neutrophils and macrophages on days 0, 1, 4, and 7. (**C**) Violin plots showing *Trem1* and *Mmp9* expression levels in myeloid cells on days 0, 1, 4, and 7; both genes are upregulated in

*Figure 5 continued on next page*

*Figure 5 continued*

neutrophils after ligature placement. (**D**) Violin plots showing *Tgfb1* and *Tnf* expression in myeloid cells on days 0, 1, 4, and 7; no obvious induction is observed in response to ligature placement. (**E**) Dot plots depicting expression levels of the C motif chemokine ligand (CCL) genes *Ccl2*, *Ccl3*, *Ccl4*, *Ccl6*, *Ccl9*. (**F**) The expression of CXC motif chemokine ligand (CXCL) genes *Cxcl2*, *Cxcl3*, *Cxcl4,* and *Cxcl9*. Chemokine expression in myeloid cells was unrelated to progression of gingival inflammation from days 1–7.

The online version of this article includes the following figure supplement(s) for figure 5:

**Figure supplement 1.** Identification of myeloid cell subpopulations in mouse gingival tissue during periodontitis development by scRNA-seq.

## The role of AG fibroblasts and neutrophils in ILC3 development in periodontitis

Based on our present data, we hypothesize that ILC3s within the gingival tissue play a pathological role in cervical alveolar bone resorption in our mouse model. We therefore aimed to identify the cells that promote ILC3 development in mice. Similar to Th17 cells, ILC3 development was shown to be triggered by IL-6 and IL-23a (*Bielecki et al., 2021*; *Powell et al., 2015*). A survey of our scRNA-seq data identified epithelial cells, fibroblasts, and myeloid cells as the source of *Il6* in gingival tissue (*Figure 9A*). On days 1 and 4 following ligature placement, AG fibroblasts primarily expressed *Il6*, whereas neutrophils became the predominant source on day 7. We further found that gingival epithelial cells comprise at least four different subpopulations, plus an additional group displaying an epithelial–mesenchymal transition (EMT) phenotype on day 7; *Il6* was expressed by several of these epithelial subpopulations, including the EMT subgroup (*Figure 9A*, *Figure 9—figure supplement 1A–D*). *Il23a* was also detected AG fibroblasts and myeloid cells (*Figure 9B*), with expression present in all epithelial cell subsets at various points in periodontitis development (*Figure 9B*, *Figure 9—figure supplement 1D*).

Lastly, we evaluated potential interactions between chemokine ligands expressed by AG fibroblasts and neutrophils and chemokine receptors in innate immune cells. Our data suggest the presence of a chemokine–receptor association between ILC3s and both AG fibroblasts (*Figure 9C*) and neutrophils (*Figure 9D*), although interactions with other innate immune cells are also possible. NicheNet analysis further identified potential ligand–target gene associations between lymphocytes, including ILC3s, and both fibroblasts (*Figure 9E*) and myeloid cells (*Figure 9F*). Ligand expression was more prominent in AG fibroblasts than in other fibroblast subpopulations (*Figure 9G*) and elevated in neutrophils relative to macrophages (*Figure 9H*). Additionally, target gene expression was detected in ILC3s, although it was not specific to these cells (*Figure 9I and J*). Thus, in total, our data suggest a regulatory role for a newly identified AG fibroblast subpopulation in the gingiva, which appears to orchestrate chronic gingival inflammation, at least in the early stages, and to promote alveolar bone resorption via stimulation of neutrophils and ILC3s.

## Discussion

This study revealed that the gene signature of a unique and previously uncharacterized subpopulation of gingival fibroblasts, referred to as AG fibroblasts, could possess the functional capability to serve as an oral immune surveillant and orchestrate the initiate gingival inflammation. Oral barrier immunity presents a complex interaction between different types of immune cells to protect and maintain the oral environment and structure. Given the frequent exposure to a diverse commensal and pathological microbiome, physical and chemical insults, and dietary and airborne antigens, the oral barrier immune mechanism must resiliently establish the highly tolerant homeostasis (*Moutsopoulos and Konkel, 2018*). This oral immune homeostasis mechanism is not yet fully understood to date; however, the aberrant oral immune response and gingival chronic inflammation leading to periodontitis have provided an important clue to elucidate the oral barrier immunity.

The previous studies suggested gingival intraepithelial γδT cells (*Chen et al., 2022*), a subset of neutrophils (*Fine et al., 2016*), macrophages (*Metcalfe et al., 2021*), and dendritic cells (*Hovav, 2014*) should serve the candidate immune surveillant in the oral barrier tissue. These immune cells recognize the pathological signals through the TLR sensing mechanism. TLR2[-/-], TLR4[-/-], TLR2&4[-/-] (*Lin et al., 2017*), and TLR9[-/-] (*Kim et al., 2015*; *Crump et al., 2017*) mice exhibited the reduced alveolar bone loss in various mouse periodontitis models, suggesting that TLRs indeed played an important

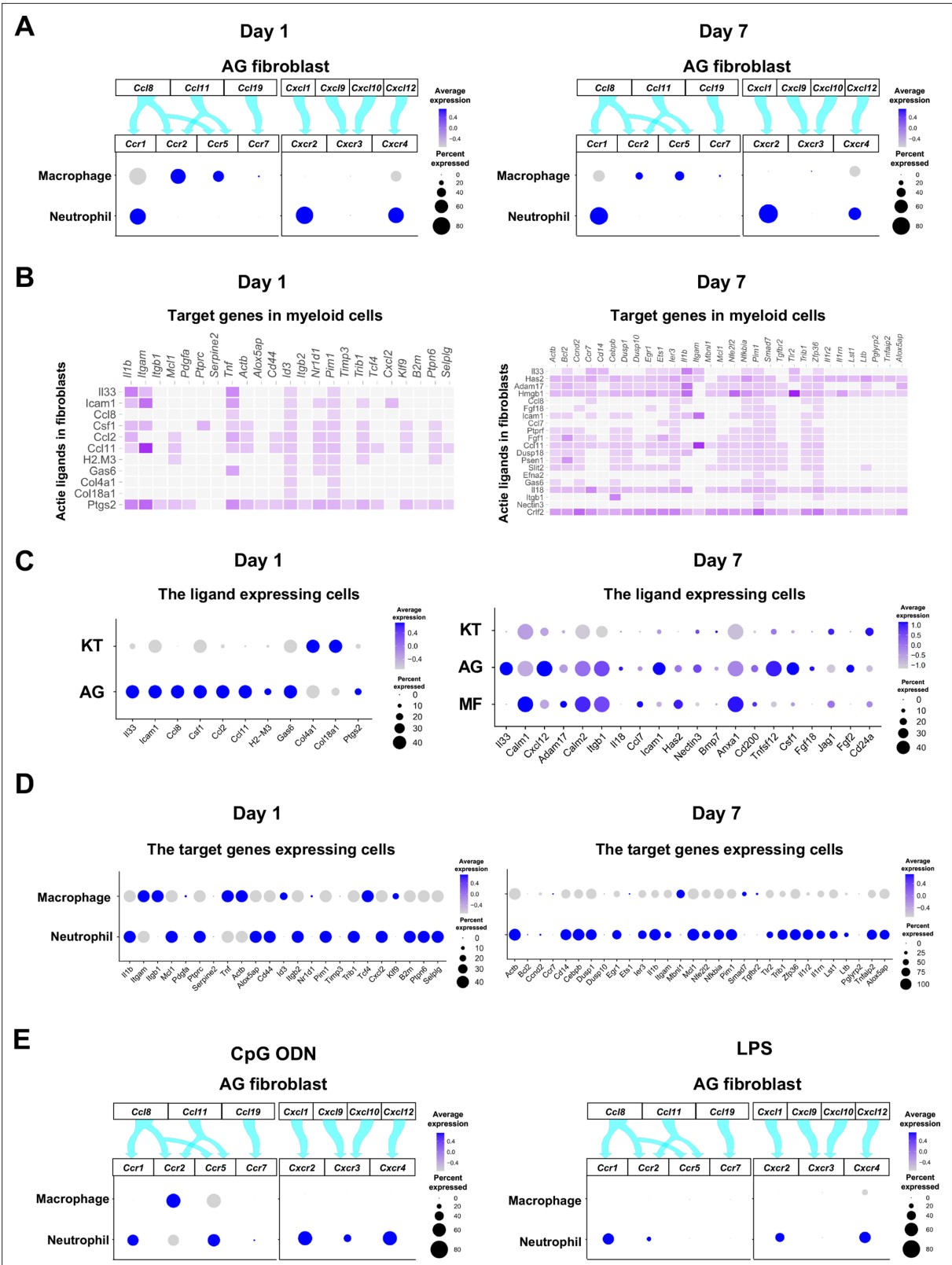

**Figure 6.** Role of AG fibroblasts in myeloid cell activation. (**A**) Interaction between chemokine ligands expressed by AG fibroblasts and their putative chemokine receptors expressed by myeloid cells during periodontitis development. Dot plots depicting expression levels of the CC chemokine receptor (CCR) and the CXC chemokine receptor (CXCR) genes *Ccr1*, *Ccr2*, *Ccr5*, *Ccr7*, *Cxcr2*, *Cxcr3*, and *Cxcr4* in myeloid cell subpopulations on days 1 and 7 following ligature placement. (**B**) NicheNet ligand–target matrix indicating the regulatory potential between active ligands expressed

*Figure 6 continued*

in fibroblasts and target genes expressed in myeloid cells from the p-EMT program on days 1 and 7. (**E**) Dot plot depicting expression levels of active ligand genes from panel (**B**) in fibroblast subpopulations on days 1 and 7. (**D**) Dot plot depicting expression levels of target genes from panel (**B**) in myeloid cell subpopulations on days 1 and 7. Results suggest a strong intercellular communication network from AG fibroblasts to neutrophils. (**E**) Interaction between chemokine ligands expressed by AG fibroblasts and their putative chemokine receptors expressed by myeloid cells in the maxillary topical application (MTA) model of cytidine phosphate guanosine oligonucleotide (CpG ODN) and lipopolysaccharide (LPS).

role in developing discordant oral barrier immunity. However, these studies did not identify the TLR-carrying surveillant cells centrally involved in initiating the chronic gingival inflammation.

It has been suggested that TLR-expressing gingival fibroblasts may regulate innate immune responses (*Qian et al., 2021*; *Naruishi, 2022*). In this study, we found that not all gingival fibroblasts acquired an immune-sensing capability, but rather, AG fibroblasts represented a distinct fibroblast subpopulation, capable of responding to microbial and tissue damage signals to serve initiate immune surveillance. To test if TLR ligands stimulate the AG fibroblast activation, we applied a newly developed mouse system employing the discrete MTA model (*Kondo et al., 2022*). This study used TLR9 ligand: CpG ODN; and TLR2/4 ligand: LPS. Topical application of these TLR ligands to the maxillary gingival tissue activated AG fibroblasts and increased expression of CC and CXC chemokines (*Figure 4F*). It was noted that the CpG ODN application did not upregulate *Tlr9* expression, while the LPS application increased *Tlr2/4* expression. *TLR9* mRNA level has not been fully correlated to the chronic inflammatory diseases; however, the pathological activation of TLR9 caused the discordant downstream inflammation (*Dragasevic et al., 2018*). In this study, the differential expression of the TLR downstream signaling molecules suggested the ligand-specific response by AG fibroblasts. In fact, both stimulants increased the expression of myeloid differentiation marker 88 (*Myd88*), IL1R-associated kinase (*Irak*), mitogen-activated protein kinase (*Map3k7*), and NF-kB subunit *RelA* (*Figure 4G*), suggesting that the corresponding TLRs were indeed stimulated (*Takeshita et al., 2001*; *Takeda and Akira, 2004*). The *Tlr* expression pattern of day 1 AG fibroblasts was similar to that of the MTA of CpG ODN, whereas the day 4 AG fibroblasts resembled the *Tlr* expression pattern of the MTA of LPS (*Figure 3B*). It is tempting to speculate that microbial DNA/TLR9 activation of AG fibroblasts may initiate the early pathological process followed by the LPS/TRL2/4 stimulation in the more established periodontal inflammation. Taken together, we hypothesize that AG fibroblasts sensitively detect the microbial signals and play a previously unrecognized role in innate immune regulation during the early periodontitis development.

One of the major observations in this study was that AG fibroblasts clearly upregulated the chemokine expression only after the pathological stimulation. CCL and CXCL chemokines orchestrate the chronic inflammation through the ligation and activation of their putative chemokine receptors in a specific mechanism. For example, neutrophils express a relatively limited number of chemokine receptors—CXCR2, CXCR4, and CCR1 (*Sabroe et al., 2005*; *Huston and Muhm, 1989*). Therefore, neutrophil migration and trafficking to the affected gingiva as well as its pathological transformation in chronic periodontitis patients (*Hajishengallis, 2020*; *Kolaczkowska and Kubes, 2013*; *Tecchio and Cassatella, 2016*) may be guided through the specific chemokine–receptor interaction. Consistent with prior studies, we detected the expression of these chemokine receptors in gingival neutrophils, and they were linked to the chemokine ligands expressed by AG fibroblasts—CXCL1, CXCL12, and CCL8, respectively (*Figure 6*). In addition, neutrophils can respond to various chemoattractants that modulate and fine-tune various cellular behaviors, such as migration direction, adhesion strength, and functional heterogeneity (*Metzemaekers et al., 2020*). In this study, we observed evidence for substantial intracellular communication between AG fibroblasts and neutrophils through ligand–target gene interactions (*Figure 6*). Although validation of each putative ligand–receptor interaction is beyond the scope of this study, our data suggest that AG fibroblasts support neutrophil trafficking in both early and established periodontitis lesions.

A primary pathological consequence of periodontitis is uncontrolled alveolar bone resorption, leading to tooth loss, which is thought to be mediated by osteoimmune pathways involving IL-17-expressing Th17 cells (*Deng et al., 2022*). Surprisingly, analysis of our scRNA-seq revealed that the expression of *Rorc*, *Il17a*, and *Il17f* was not detected in CD4[+] Th cells, but rather, in lymphocytes with ILC characteristics (*Figure 8*). ILCs share phenotypic and functional features with CD4[+] T cells, although they lack antigen-specific T cell receptors (*Eberl et al., 2015*). The gingival ILCs expressed

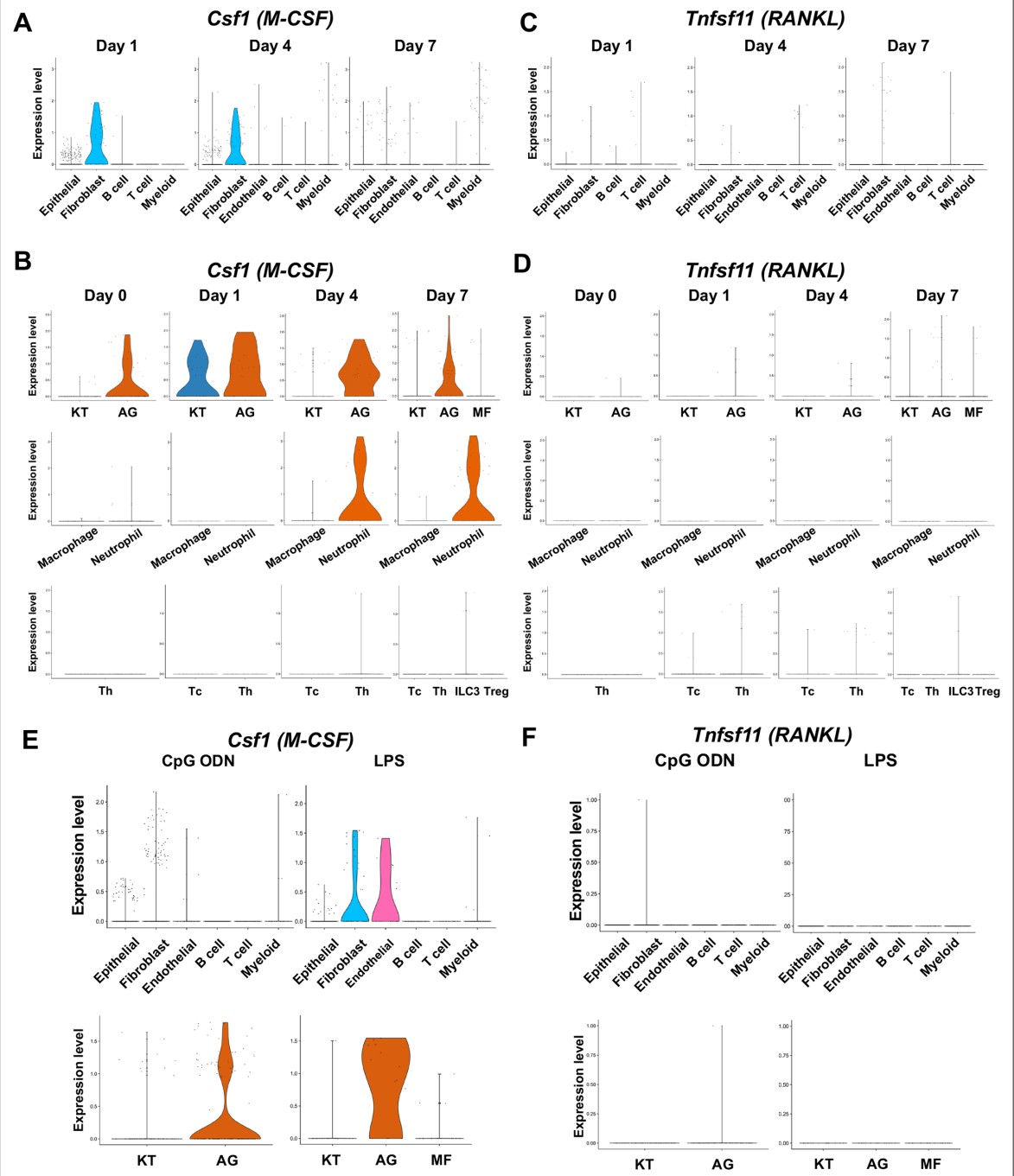

**Figure 7.** The cellular source of osteoclastogenic cytokines: macrophage-colony stimulating factor (M-CSF) and receptor activator of nuclear factor kappa- B ligand (RANKL) in the ligature-induced model and the maxillary topical application (MTA) model. (**A**) Violin plots showing expression levels of the M-CSF-encoding gene, *Csf1* (M-CSF), in each major cell type from the ligature-induced periodontitis model. (**B**) Violin plots showing expression levels of *Csf1* in fibroblast subpopulations, myeloid cell subpopulations, and T cell subpopulations. (**C**) Violin plots showing expression levels of the RANKL-encoding gene, *Tnfsf11*, in each major cell type. (**D**) Violin plots showing expression levels of *Tnfsf11* in fibroblast subpopulations, myeloid cell subpopulations, and T cell subpopulations. (**E**) Violin plots of *Csf1*-expressing cells in the MTA model. (**F**) Violin plots of *Tnfsf11*-expressing cells in the MTA model. AG fibroblasts predominantly expressed *Tnfsf11*.

*Rorc*, *Il17a*, and *Il17f*, but not *Tbx21* and *Gata3*, indicating that they were ILC3s. Notably, recent clinical studies have reported the presence of ILCs in gingiva from human periodontitis patients (***Brown et al., 2018***; ***Dutzan et al., 2016***) and leptin receptor-deficient mice (***Wang et al., 2022***), although their pathological contributions have not been fully elucidated.

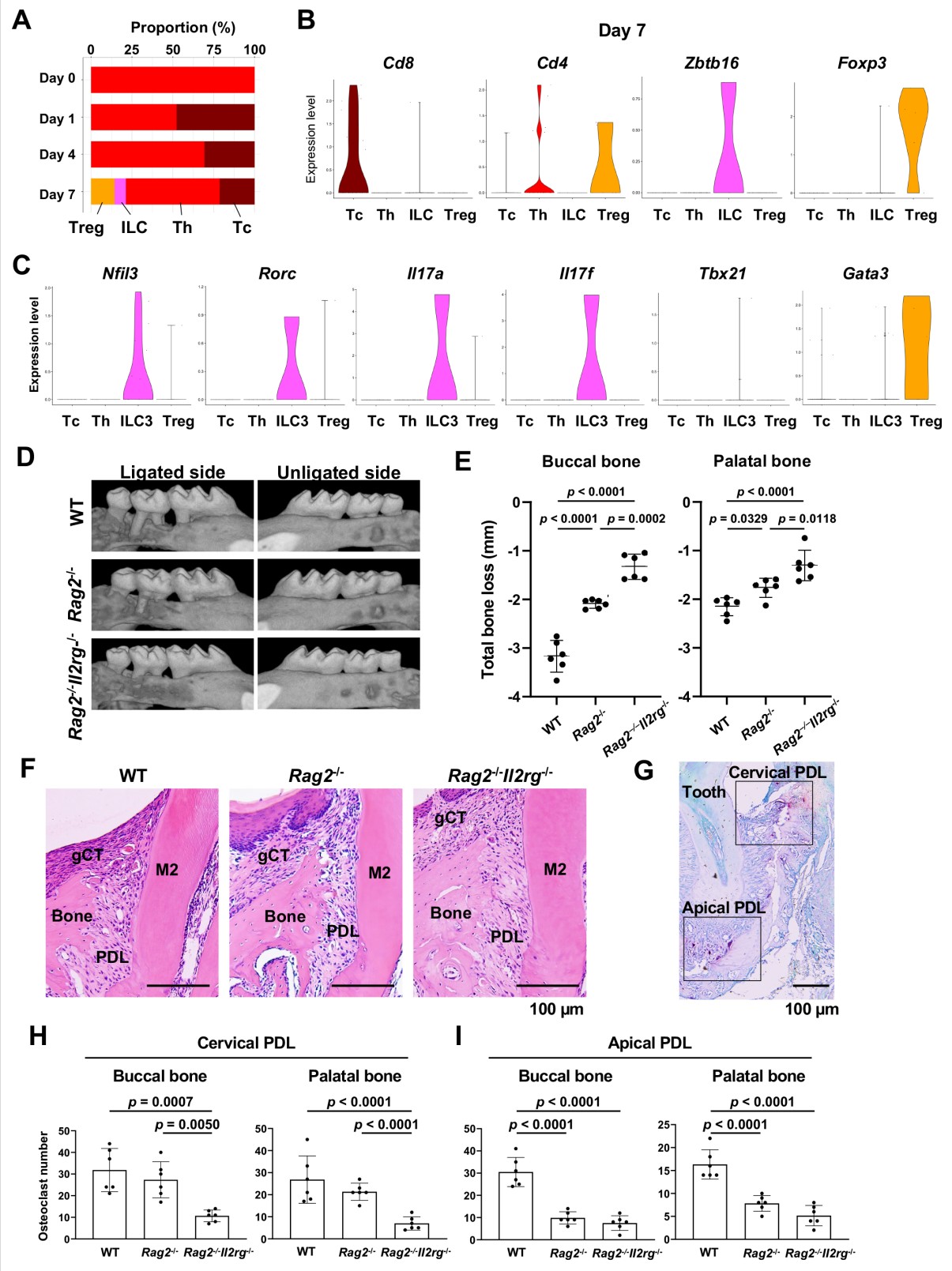

**Figure 8.** Type 3 innate lymphoid cells (ILC3s) are critical for cervical alveolar bone resorption in the ligature-induced periodontitis development. (**A**) Proportion plots showing the relative amounts of T cell subpopulations in gingival tissue during periodontitis development. Treg, T regulatory cells; ILC, innate lymphoid cells; Th, T helper cells; Tc, cytotoxic T cells. (**B**) Violin plots showing expression levels of the T cell marker genes *Cd8* (Tc), *Cd4* (Th), *Zbtb16* (ILC), and *Foxp3* (Treg) on day 7 following ligature placement. (**C**) Violin plots showing expression levels of *Nfil3*, *Rorγ*, *Il17a*, *Il17f*, *Tbx21*, and

*Figure 8 continued on next page*

*Figure 8 continued*

*Gata3* on day 7 following ligature placement. These gene signatures indicate that gingival ILCs primarily comprise ILC3s. (**D**) Representative micro-computed tomography (microCT) images of the maxilla taken from the lateral view for the ligated side and from the contralateral view for the unligated side. (**E**) Alveolar bone loss was determined from the total distance between the cementoenamel junction (CEJ) and the alveolar bone crest (ABC) of the buccal bone or palatal bone at six sites in the ligated side (n = 6). (**F**) HE staining of the periodontal tissue on day 7. gCT, gingival connective tissue; Bone, alveolar bone; PDL, periodontal ligament; scale bars, 100 µm. (**G**) Tartrate-resistant acid phosphatase (TRAP) staining of periodontal tissue from WT mice on day 7; scale bar, 100 µm. Total number of TRAP-positive cells in a 0.01 mm$^2$ area of the buccal and palatal bone in the cervical PDL site (**H**) and apical PDL site (**I**) (n = 6). Significance was determined by ANOVA, with Tukey's multiple-comparison test. Data are presented as mean values ± SD; p<0.05 was considered significant.

The online version of this article includes the following figure supplement(s) for figure 8:

**Figure supplement 1.** Effects of innate lymphoid cell (ILC) deletion on alveolar bone loss in the mouse periodontitis model.

**Figure supplement 2.** Human periodontitis phenotype.

In this study, we further explored the role of ILC3s in periodontitis-associated bone loss using *Rag2*$^{-/-}$ and *Rag2*$^{-/-}$*Il2rg*$^{-/-}$ mice. *Rag2*$^{-/-}$ mutation prevents V(D)J recombination required for generating immunoglobulin and T cell receptors, resulting in the production of functionally immature B and T cells, including Th17 cells. However, ILCs do not undergo genomic receptor rearrangements and, thus, are unaffected by *Rag2*$^{-/-}$ mutation. In contrast, *Rag2*$^{-/-}$*Il2rg*$^{-/-}$ mice have the additional *Il2rg*$^{-/-}$ mutation, which disables common γ chain cytokines (γc). Therefore, in addition to non-functional B and T cells, these animals also have defective γc-dependent ILCs. The ligature-placed *Rag2*$^{-/-}$ mice exhibited a reduced number of osteoclasts in the apical PDL area, indicative of reduced bone resorption in this root apical region. By contrast, *Rag2*$^{-/-}$*Il2rg*$^{-/-}$ mice demonstrated a significant loss of bone resorption in both the cervical and apical PDL areas (***Figure 8***). Human periodontitis induces the localized alveolar bone resorption at the cervical PDL zone interfacing the gingival inflammatory legion. In contrast, apical alveolar bone resorption is observed in clinical cases of root canal infection (apical periodontitis). Therefore, our findings suggest that ILC3s, which are differentially impacted by *Rag2*$^{-/-}$*Il2rg*$^{-/-}$ and *Rag2*$^{-/-}$ mutations, may be primarily responsible for human periodontitis-like cervical alveolar bone resorption near the site of gingival inflammation.

Differentiation of ILC3s and Th17 cells is mediated by similar environmental cues, with IL-6 and IL-23a playing critical roles in both pathways (***Klose et al., 2013***; ***Sawa et al., 2010***). Our present scRNA-seq data (***Figure 9***) suggest that AG fibroblasts and neutrophils are the primary cellular sources of *Il6* during the early and later stages, respectively, of periodontitis development. In contrast, *Il23a* was found to be expressed by a variety of cell types, such as AG fibroblasts, myeloid cells, and multiple subsets of epithelial cells, including those with an EMT phenotype (***Wadie et al., 2021***). Further, CC and CXC chemokine–receptor associations between ILC3s and both AG fibroblasts and neutrophils appeared to be non-specific. Therefore, our data suggest that rather than a specific trigger, the collective gingival environment, which includes AG fibroblasts, might contribute to ILC3 differentiation.

In conclusion, based on our present findings, we propose that a previously unrecognized AG fibroblast subpopulation in the gingiva can facilitate immune surveillance and participate in the pathological regulation of innate immune cells, such as proinflammatory neutrophils, within oral barrier tissue (***Figure 10***). Moreover, we hypothesize that ILC3s in the inflamed gingiva play a critical role in pathological alveolar bone resorption in the mouse model of periodontitis and, potentially, in human disease. Thus, the newly proposed AG fibroblast–neutrophil–ILC3 axis may hold valuable clues for unraveling the pathological mechanisms underlying periodontitis development. Moreover, these findings also provide a basis for investigation of new preventive and therapeutic strategies to contain oral barrier inflammation and potentially sever the link between periodontitis and debilitating non-communicative metabolic and cardiovascular diseases.

## Materials and methods
### Animal care

All protocols for animal experiments were reviewed and approved by the University of California Los Angeles (UCLA) Animal Research Committee (ARC# 2003-009) and followed the Public Health Service Policy for the Humane Care and Use of Laboratory Animals and the UCLA Animal Care and Use Training Manual guidelines. C57BL/6J WT (Strain # 000664; RRID:IMSR_JAX:000664), *Rag2*$^{-/-}$ (Strain #

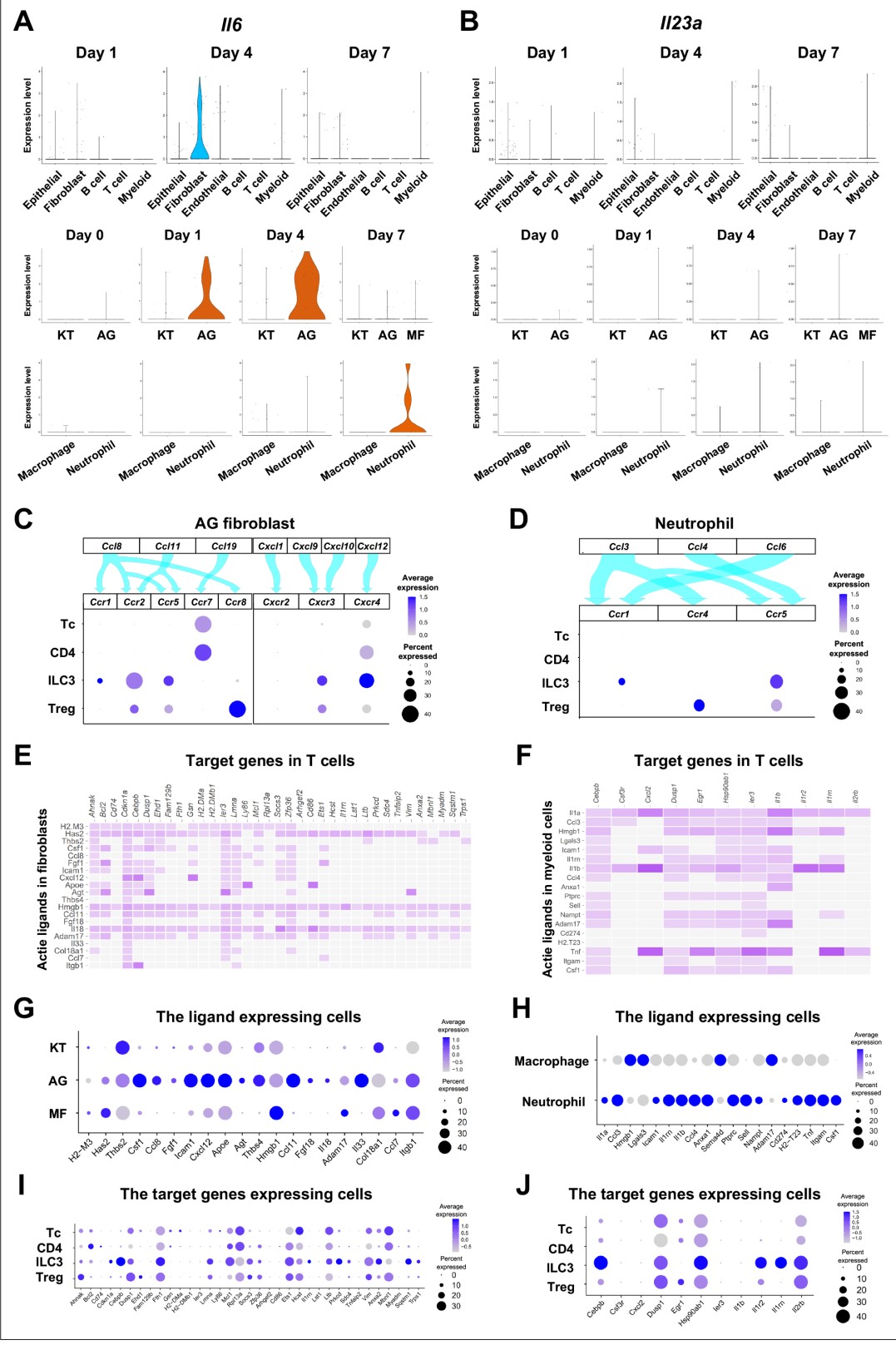

**Figure 9.** The role of AG fibroblasts and neutrophils in type 3 innate lymphoid cell (ILC3) development in periodontitis. Violin plots showing expression levels of the genes encoding interleukin (IL)-6 (*Il6*) (**A**) and IL-23 (*Il23a*) (**B**) in each major cell type, fibroblast subpopulations, and myeloid cell subpopulations during periodontitis development. (**C**) Interaction between chemokine ligands strongly expressed by AG fibroblasts and their putative

*Figure 9 continued on next page*

*Figure 9 continued*

chemokine receptors expressed by T cells, including ILCs. Dot plots depict gene expression levels of *Ccr1*, *Ccr2*, *Ccr5*, *Ccr7*, *Ccr8*, *Cxcr2*, *Cxcr3*, and *Cxcr4* in T cell subpopulations on day 7 following ligature placement. (D) Interaction between chemokine ligands strongly expressed by neutrophils and their putative chemokine receptors expressed by T cells. Dot plots depicting gene expression levels of *Ccr1*, *Ccr4*, and *Ccr5* in T cell subpopulations on day 7 following ligature placement. (E) NicheNet ligand–target matrix denoting the regulatory potential between active ligands in fibroblasts and target genes in T cells from the p-EMT program on day 7 following ligature placement. (F) NicheNet ligand–target matrix denoting the regulatory potential between active ligands in myeloid cells and target genes in T cells from the p-EMT program on day 7 following ligature placement. (G) Dot plot depicting expression levels of active ligand genes from panel (E) in fibroblast subpopulations. (H) Dot plot depicting expression levels of active ligand genes from panel (F) in myeloid cell subpopulations. (I) Dot plot depicting expression levels of target genes from pane (E) in T cell subpopulations. (J) Dot plot depicting expression levels of target genes from panel (F) in T cell subpopulations.

The online version of this article includes the following figure supplement(s) for figure 9:

**Figure supplement 1.** Epithelial cell subpopulations involved in periodontitis development in mice.

008449; RRID:IMSR_JAX:008449), and *Rag2^{-/-}Il2rg^{-/-}* (Strain # 014593; RRID:IMSR_JAX:014593) mice were purchased from the Jackson Laboratory (Bar Harbor, ME). Animals had free access to regular rodent diet and water and were maintained in standard housing conditions with 12 hr light/dark cycles in the Division of Laboratory Animal Medicine at UCLA. All animal experiments were conducted according to the guideline of the Animal Research; Reporting of In Vivo Experiments (ARRIVE: Essential 10).

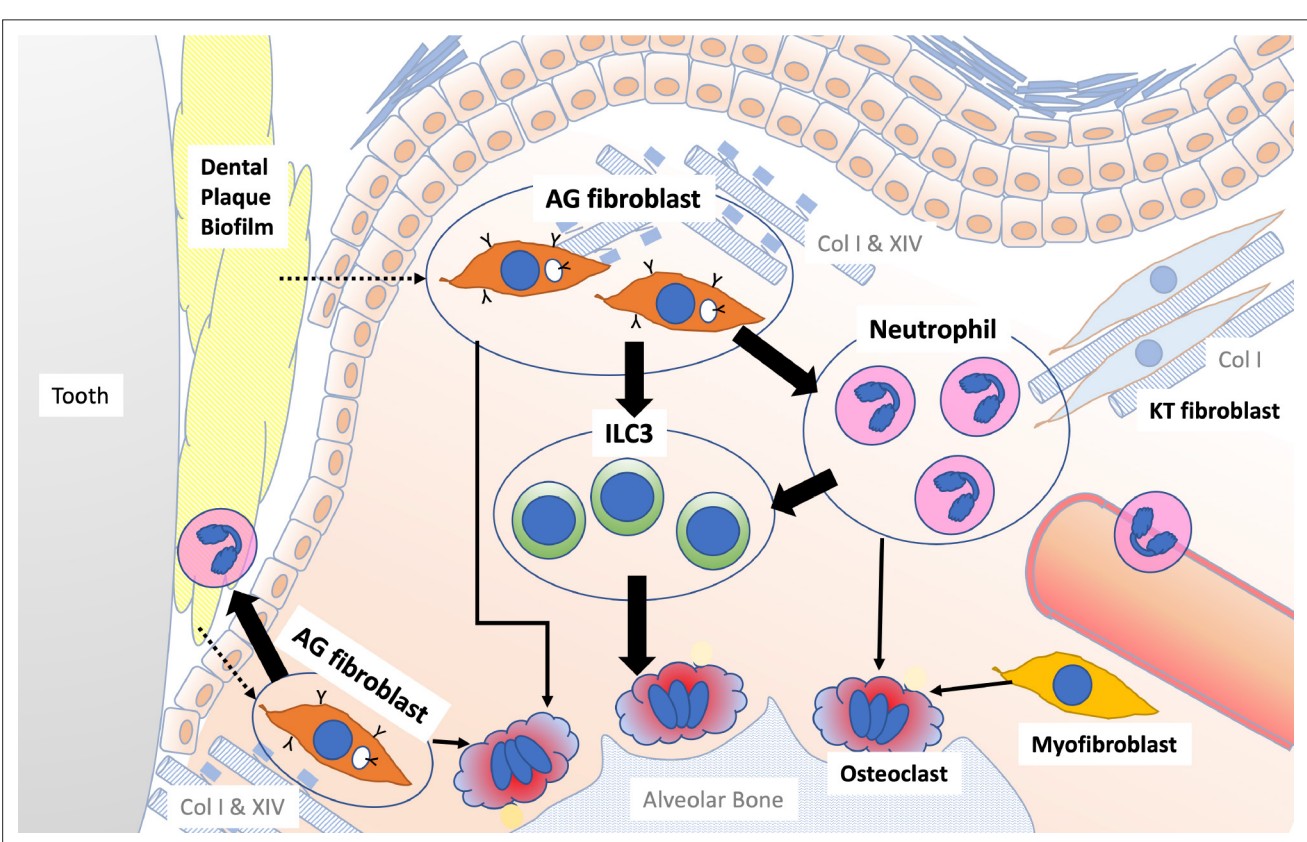

**Figure 10.** Schematic overview of the newly proposed AG fibroblast–neutrophil–ILC3 axis. We propose that periodontal inflammation is initiated by the activation of AG fibroblasts, which secrete chemokines and cytokines that recruit neutrophils to sites of tissue damage. Activated neutrophils and AG fibroblasts, in turn, activate ILC3s, leading to the production of proinflammatory IL-17 cytokines. Ultimately, cervical alveolar bone resorption is facilitated by a localized osteoclastogenic environment, induced by activated ILC3s, together with AG fibroblasts, neutrophils, myofibroblasts, and gingival epithelial cells, including those with an epithelial–mesenchymal transition (EMT) phenotype.

## Evaluation of gingival defect and alveolar bone resorption in a ligature-induced mouse model of periodontitis

A silk thread was gently tied around the left maxillary second molar of 8- to 12-week-old female WT, $Rag2^{-/-}$, and $Rag2^{-/-}Il2rg^{-/-}$ mice under general inhalation anesthesia with isoflurane (Henry Schein, Melville, NY) following the published protocol (*Abe and Hajishengallis, 2013*; PMCID: PMC3707981; DOI: https://doi.org/10.1016/j.jim.2013.05.002).

WT mice were randomly chosen and euthanized by 100% $CO_2$ inhalation on days 1, 3, and 7 after ligature placement (n = 6 per time point). WT mice without ligature placement were used as day 0 pre-periodontitis control (n = 6). The ligature-induced mouse model using WT mice was replicated at least two times in our laboratory.

$Rag2^{-/-}$ and $Rag2^{-/-}Il2rg^{-/-}$ mice were euthanized on day 7 (n = 6) and a separate set of WT mice were used as a control group and euthanized on day 7 (n = 6) in this experiment.

The palatal tissue was digitally photographed, and maxillae were harvested and fixed in 10% buffered formalin (Thermo Fisher Scientific, Waltham, MA). The gingival defect area was measured from digital photographs using the ImageJ Java-based image-processing program (NIH, Bethesda, MD) and normalized to the circumferential area of the maxillary first molar. Fixed maxillae were subjected to microCT imaging at an energy level of 60 kV and 166 μA, and 3D images were reconstructed from microCT scans (Skyscan 1275: Bruker, Billerica, MA). Alveolar bone loss was assessed at three sites (mesiobuccal cusp, distobuccal cusp, and distal cusp) of the first molar, two sites (mesiobuccal cusp and distobuccal cusp) of the second molar, and one site (buccal cusp) of the third molar by measuring the distance from the cementoenamel junction (CEJ) to the alveolar bone crest (ABC) on the buccal and palatal side of the alveolar bone. Total bone loss was calculated from the six-site total CEJ–ABC distance. The bone volume/total volume (BV/TV) ratio, bone surface, trabecular number (Tb.N), and trabecular thickness (Tb.Th) in the buccal side of alveolar bone of the second molar were determined using the proprietary analysis program (CTan: Bruker).

## Evaluation of gingival effect in the MTA model

We have developed a method to apply chemical therapeutic agents topically to the mouse maxillary tissue (*Okawa et al., 2022a*; *Okawa et al., 2022b*). We used the MTA model to apply oral microbial components (*Kondo et al., 2022*) (PMCID: PMC9474870; DOI: 10.1038/s42003-022-03896-7). This study used the MTA model using 1 μg/ml unmethylated CpG oligonucleotide (CpG ODN: InvivoGen, San Diego, CA) or *P. gingivalis* lipopolysaccharide (LPS: InvivoGen). First, a custom-made oral appliance was fabricated using clear dental resin, covering the maxillary/palatal tissue between the molars. Mice were anesthetized and placed on a supine position. CpG ODN (3 μl) or LPS (3 μl) was pipetted over the maxillary tissue and the oral appliance was placed to hold the solution with a bite block for 1 hr in an anesthetization chamber. Mice were then transferred to the operation table and the oral appliance and bite block were removed. In general, there was no remaining solution in the mouth. Mice were returned to the cage in the vivarium. On day 4 of the MTA, mice were euthanized by 100% $CO_2$ inhalation and the maxillary tissue was harvested and fixed with 10% buffered formalin for histological evaluation.

## Histological analysis

Fixed maxillae were decalcified in 10% EDTA (Sigma-Aldrich) for 3 wk and then embedded in paraffin. Histological cross-sections were stained with hematoxylin and eosin (HE) and evaluated on a light microscope. Adjacent paraffin sections (4 μm) were subjected to a heat-induced epitope retrieval procedure and then immunohistochemically stained with polyclonal antibodies to CD45 (#PA5-11671, Thermo Fisher Scientific), COL14A1 (#PA5-49916, Thermo Fisher Scientific), CXCL12 (#PA5-30603, Thermo Fisher Scientific), or Ctsk (PA5-14270, Thermo Fisher Scientific) at the suggested dilution, followed by secondary antibody application, diaminobenzidine staining, and methylene blue counterstaining.

Using maxillary cross-sections of WT, Rag2$^{-/-}$, and $Rag2^{-/-}Il2rg^{-/-}$ mice, osteoclasts were evaluated by TRAP staining using a commercially available kit (Acid Phosphatase TRAP kit, Sigma-Aldrich), according to the manufacturer's instructions. TRAP-positive cells were counted in the 0.01 mm$^2$ area.

### Single-cell dissociation from mouse maxillary gingiva

On days 1, 4, and 7 after ligature placement, and on day 4 of the MTA of CpG ODN or LPS, mice were euthanized by 100% $CO_2$ inhalation. Maxillary gingival tissues (n = 4 per group) were harvested from freshly isolated mouse maxillae.

## Collagenase II treatment

The tissues were cut into 1 mm pieces and placed immediately into digestion buffer, containing 1 mg/ml collagenase II (Life Technologies, Thermo Fisher Scientific), 10 units/ml DNase I (Sigma-Aldrich, St. Louis, MO), and 1% bovine serum albumin (BSA; Sigma-Aldrich) in Dulbecco's Modified Eagle Medium (DMEM; Life Technologies). The tissues were incubated in digestion buffer for 20 min at 37°C on a shaker at 150 rpm and then passed through a 70 μm cell strainer. The collected cells were pelleted at 1500 rpm for 10 min at 4°C and resuspended in phosphate-buffered saline (PBS; Life Technologies), supplemented with 0.04% BSA (cell suspension A).

## Trypsin treatment

Immediately following collagenase II treatment, tissues were incubated in 0.25% trypsin (Life Technologies) and 10 units/ml DNase I for 30 min at 37°C on a 150 rpm shaker. Trypsin was neutralized with fetal bovine serum (Life Technologies), and the tissues were passed through a 70 μm cell strainer and washed with DMEM. The collected cells were then pelleted at 1500 rpm for 10 min at 4°C and resuspended in PBS with 0.04% BSA (cell suspension B). Cell suspensions A and B were combined in one tube. An equal number of combined cell suspensions A and B from four animals per group were combined for scRNA-seq analysis (10X Genomics, San Francisco, CA).

### Cell clustering and Identification

Cell Ranger was used to align reads, generate feature–barcode matrices, and perform clustering and gene expression analyses on the scRNA-seq data, and the output from this program was analyzed using the R-program Seurat (https://satijalab.org/seurat/). Cells with <2400 genes detected or >1% mitochondrial gene expression were filtered out as low-quality cells. Individual gene counts for each cell were divided by the total gene counts for that cell and multiplied by a scale factor of 10,000; natural-log transformation was then applied to the counts. The FindVariableFeatures function was used to select 2000 variable genes with default parameters, and the ScaleData function was used to scale and center the counts in the dataset. Principal component analysis and Uniform Manifold Approximation and Projection dimensional reduction were performed on variably expressed genes. The cluster markers were found using the FindAllMarkers function, and cell types were manually annotated based on the cluster markers. Cell types were assigned based on expression of cell marker genes, and gene expression within different cell types was displayed using dot plots and violin plots.

### Functional annotation and pathway enrichment analysis

Annotation and visualization of GO terms were performed by Metascape (http://metascape.org/gp/index.html#/main/step1). The top 100 differentially expressed genes in each population were input and filtered with the term 'immune.' Filtered genes were then input, and only 'biological process' gene sets were retrieved from the GO database.

### Ligand–target matrix prediction

NicheNet (v.1.0.0, https://github.com/saeyslab/nichenetr; *Browaeys et al., 2020*; *Saeys Lab, 2023*) was used to predict interactions between cell types. In brief, the integrated Seurat object containing each cell subpopulation was input into the NicheNet Seurat wrapper. Sender cells and receiver cells were determined, and interactions between active ligands expressed by sender cells and target receptors expressed by receiver cells were predicted based on information in signaling and ligand–receptor databases.

## Statistical analysis

Statistical analysis to assess differences among multiple experimental groups was performed using one-way ANOVA with Tukey's multiple-comparison test; $p < 0.05$ was considered to be statistically significant.

## Acknowledgements

We thank Dr. Connie Lee of Division of Periodontology at the UCLA School of Dentistry for her guidance on the clinical manifestations of human periodontitis. We also thank Dr. Yunfeng Li of the Translational Pathology Core Laboratory, Department of Pathology and Laboratory Medicine, David Geffen School of Medicine at UCLA for her immunohistology work. This study was supported by NIH grants R01DE022550, R44DE025524, and by SINTX Technologies. This investigation was performed in part in the research facility constructed using C06RR014529.

## Additional information

### Competing interests

Akishige Hokugo: A.H. received a research fund from Maruho Co. Ltd. Ichiro Nishimura: I.N. is a consultant for FUJI FILM Corp and BioVinc LLC and received a research fund from SINTX Technologies, Inc I.N. received a research fund from Maruho Co. Ltd. The other authors declare that no competing interests exist.

### Funding

| Funder | Grant reference number | Author |
| --- | --- | --- |
| National Institute of Dental and Craniofacial Research | R01DE022550 | Ichiro Nishimura |
| National Institute of Dental and Craniofacial Research | R44DE025524 | Ichiro Nishimura |
| SINTX Technologies | | Ichiro Nishimura |

The funders had no role in study design, data collection and interpretation, or the decision to submit the work for publication.

### Author contributions

Takeru Kondo, Data curation, Formal analysis, Validation, Investigation, Methodology, Writing – original draft, Writing – review and editing; Annie Gleason, Software, Formal analysis, Visualization, Methodology, Writing – review and editing; Hiroko Okawa, Data curation, Investigation, Methodology, Writing – review and editing; Akishige Hokugo, Investigation, Methodology, Writing – review and editing; Ichiro Nishimura, Conceptualization, Formal analysis, Supervision, Funding acquisition, Validation, Writing – original draft, Project administration, Writing – review and editing

### Author ORCIDs

Takeru Kondo (ORCID) http://orcid.org/0009-0002-1560-1741
Akishige Hokugo (ORCID) https://orcid.org/0000-0002-7097-3364
Ichiro Nishimura (ORCID) https://orcid.org/0000-0002-3749-9445

### Ethics

All protocols for animal experiments were reviewed and approved by the University of California Los Angeles (UCLA) Animal Research Committee (ARC# 2003-009) and followed the Public Health Service Policy for the Humane Care and Use of Laboratory Animals and the UCLA Animal Care and Use Training Manual guidelines. C57BL/6J WT, Rag2-/-, and Rag2-/-Il2rg-/- mice were purchased from the Jackson Laboratory (Bar Harbor, ME, USA). Animals had free access to regular rodent diet and water and were maintained in standard housing conditions with 12-h light/dark cycles in the Division of Laboratory Animal Medicine at UCLA.

Reviewer #1 (Public Review): https://doi.org/10.7554/eLife.88183.3.sa1
Reviewer #2 (Public Review): https://doi.org/10.7554/eLife.88183.3.sa2
Author Response https://doi.org/10.7554/eLife.88183.3.sa3

## Additional files

### Supplementary files
• MDAR checklist

### Data availability

All data generated and analyzed during this study are included in the manuscript and the Source Data file is provided in Dryad: https://doi.org/10.5061/dryad.hqbzkh1pb. Single-cell RNA-sequencing data obtained in this study are provided in NIH Gene Expression Omnibus (GSE228635): https://www.ncbi.nlm.nih.gov/geo/query/acc.cgi?acc=GSE228635.

The following datasets were generated:

| Author(s) | Year | Dataset title | Dataset URL | Database and Identifier |
|---|---|---|---|---|
| Nishimura I, Kondo T | 2023 | Single Cell RNA sequencing of mouse periodontitis gingiva | https://www.ncbi.nlm.nih.gov/geo/query/acc.cgi?acc=GSE228635 | NCBI Gene Expression Omnibus, GSE228635 |
| Kondo T, Gleason A, Okawa H, Hokugo A, Nishimura I | 2023 | Data from: Mouse gingival single cell transcriptomic atlas identified a novel fibroblast subpopulation activated to guide oral barrier immunity in periodontitis | https://doi.org/10.5061/dryad.hqbzkh1pb | Dryad Digital Repository, 10.5061/dryad.hqbzkh1pb |

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
