## [Editor Report · eLife assessment]

The findings of this article provide **valuable** information on the changes of cell clusters induced by chronic periodontitis. The observation of a new fibroblast subpopulation, named AG fibroblasts, is interesting, and the strength of evidence presented is **solid**.

---

## [Referee Report · Reviewer #1 (Public Review)]

In this article, the authors found a distinct fibroblast subpopulation named AG fibroblasts, which are capable of regulating myeloid cells, T cells and ILCs, and proposed that AG fibroblasts function as a previously unrecognized surveillant to orchestrate chronic gingival inflammation in periodontitis. Generally speaking, this article is innovative and interesting.

---

## [Referee Report · Reviewer #2 (Public Review)]

This study proposed the AG fibroblast-neutrophil-ILC3 axis as a mechanism contributing to pathological inflammation in periodontitis. In this study single-cell transcriptomic analysis was performed. But the signal mechanism behind them was not evaluated.

The authors achieved their aims, and the results partially support their conclusions.

The mouse ligatured periodontitis models differ from clinical periodontitis in human, this study supplies the basis for future research in human.

---

## [Author Response]

The following is the authors’ response to the current reviews.

eLife assessmentThe findings of this article provide valuable information on the changes of cell clusters induced by chronic periodontitis. The observation of a new fibroblast subpopulation, named AG fibroblasts, is interesting, and the strength of evidence presented is solid.

We thank the Reviewing Editor and the Senior Editor for the positive assessment and strong support for our study.

**Public Reviews:**

**Reviewer #1 (Public Review):**
In this article, the authors found a distinct fibroblast subpopulation named AG fibroblasts, which are capable of regulating myeloid cells, T cells and ILCs, and proposed that AG fibroblasts function as a previously unrecognized surveillant to orchestrate chronic gingival inflammation in periodontitis. Generally speaking, this article is innovative and interesting.

We truly appreciate this public review.

**Reviewer #2 (Public Review):**
This study proposed the AG fibroblast-neutrophil-ILC3 axis as a mechanism contributing to pathological inflammation in periodontitis. In this study single-cell transcriptomic analysis was performed. But the signal mechanism behind them was not evaluated.The authors achieved their aims, and the results partially support their conclusions.

We agree that we must conduct future studies to evaluate our hypothesis.

The mouse ligatured periodontitis models differ from clinical periodontitis in human, this study supplies the basis for future research in human.

This is an important subject. We have previously expressed a concern on the mouse ligature model that the microbial composition of the mouse ligature did not mirror the human oral microbial composition. Therefore, we developed the maxillary topical application (MTA) model, in which human oral biofilm was directly applied to the maxillary gingiva. In this study, the newly developed MTA model was further dissected by single cell RNA seq, which revealed that the extracellular substances of human oral biofilm might be an important trigger of gingival inflammation. RESULT has been revised.

**Recommendations for the authors:**

**Reviewer #1 (Recommendations For The Authors):**
I appreciate the authors' efforts. I think it would be much better to simplify INTRODUCTION.

INTRODUCTION has been simplified as suggested.

**Reviewer #2 (Recommendations For The Authors):**
1. Many host cells participate in immune responses, such as gingival epithelial cells. AG fibroblast is not the only cell involved in the immune response, and the weight of its role needs to be clarified. So the expression in the conclusion should be appropriate.

RESPONSE: We agree with this comment. Our study identified the AG fibroblast–neutrophil–ILC3 axis as a previously unrecognized mechanism which could play an additional role in the complex interplay between oral barrier immune cells.

1. The main results should be included in the Abstract.

Abstract has been revised.

The following is the authors’ response to the original reviews.

We thank all reviewers for constructive critiques. We plan to perform new experiments and revise our manuscript accordingly. The text and Figures are currently undergoing the revision process. Below highlights our revision plan.

eLife assessmentThe findings of this article provide valuable information on the changes of cell clusters induced by chronic periodontitis. The observation of a new fibroblast subpopulation, which was named as AG fibroblasts, was quite interesting, but needs further evidence. The strength of evidence presented is incomplete.

We discovered a new subpopulation of gingival fibroblasts, named AG fibroblasts, using non-biased single cell RNA sequencing (scRNA-seq) of mouse gingival samples undergoing the development of ligature-induced periodontitis. AG fibroblasts exhibited a unique gene expression profile: [1] constitutive expression of type XIV collagen; and [2] ligatureinduced upregulation of Toll-Like Receptors and their downstream signals as well as chemokines such as CXCL12. Thus, we have hypothesized that AG fibroblasts initially sense the pathological stress including oral microbial stimuli and secrete inflammatory signals through chemokine expression.

The current manuscript examined the relationship between AG fibroblasts and oral barrier immune cells focusing on the chemokines and other ligands derived from AG fibroblasts and their putative receptors in those immune cells. Using scRNA-seq data mining programs, our data demonstrated the compelling evidence that AG fibroblasts should play a critical role in orchestrating the oral barrier immunity, at least at the early stages of periodontal inflammation.

We agree that it is important to explore the functional/pathological role of AG fibroblasts. In this revision, we further investigated the role of TLRs in the pathogen sensing mechanism of AG fibroblasts. To accomplish this goal, we applied a newly developed mouse model in which mice were exposed to the maxillary topical application (MTA) of oral microbial pathogens without the ligature placement. With 1 hr exposure with human oral biofilm, not with planktonic microbiota, the mice maxillary tissue exhibited measurable degradation as evidenced by the activation of cathepsin K. To dissect the role of TLRs, we applied the putative stimulants of TLR9 and TLR2/4 using the discrete MTA model. The scRNA-seq from the MTA model revealed that the application of unmethylated CpG oligonucleotide and P. gingivalis lipopolysaccharide (LPS), respectively, induced the activation of chemokines by AG fibroblast.

The revised manuscript reported this critical data with the detailed information. As such the additional figures and corresponding results, discussion and materials & methods were included.

**Public Reviews:**

**Reviewer #1 (Public Review):**
In this article, the authors found a distinct fibroblast subpopulation named AG fibroblasts, which are capable of regulating myeloid cells, T cells and ILCs, and proposed that AG fibroblasts function as a previously unrecognized surveillant to orchestrate chronic gingival inflammation in periodontitis. Generally speaking, this article is innovative and interesting, however, there are some problems that need to be addressed to improve the quality of the manuscript.

We appreciate this comment. As suggested, we further investigated the surveillant function of AG fibroblasts by reanalyzing the scRNA-seq data for stress sensing receptors such as Toll-Like Receptors (TLR). In the revision, we addressed the role of TLR in the activation of AG fibroblasts using a newly developed mouse model employing the maxillary topical application (MTA) of putative TLR stimulants. The new information clearly demonstrated that AG fibroblasts play a pivotal role as the surveillant and translating the pathogenic stimulants to oral barrier inflammation through chemokine expression.

**Reviewer #2 (Public Review):**
This study proposed the AG fibroblast-neutrophil-ILC3 axis as a mechanism contributing to pathological inflammation in periodontitis. However, the immune response in the vivo is very complex. It is difficult to determine which is the cause and which is the result. This study explores the relevant issue from one dimension, which is of great significance for a deeper understanding of the pathogenesis of periodontitis. It should be fully discussed.

We appreciate this comment. We expanded the current understanding of oral immune signal communication in Discussion and highlight how AG fibroblast may fit to it. To address this question, we expanded our investigation in the pathological signal detection by AG fibroblasts by employing the newly developed maxillary topical application (MTA) model. The revised manuscript contains the new information and expanded the discussion in the context of complex immune response.

**Reviewer #1 (Recommendations For The Authors):**
Detailed comments are listed below:Abstract：I am confused about the expression of "human periodontitis-like phenotype". How does the authors define this concept? Periodontitis is a complex disease, despite that alveolar bone resorption is a typical manifestation of periodontitis, its characteristics remain to be further studied. I hope the authors can provide some detailed information about this concept or describe it in another way.

This is an important comment. Radiographically, human periodontitis is diagnosed by alveolar bone resorption from the cervical region, not from root apex. To highlight this, we present dental radiographs of human periodontitis as supplementary information. However, we agree with this comment, our statement should be limited to alveolar bone resorption pattern in Rag2KO and Rag2gcKO mice. Abstract be revised accordingly.

Introduction:It is recommended to simplify the first to third paragraphs, and briefly explain the functions of various types of cells in different stages of periodontitis, as well as the role of different cluster markers play across the time course of periodontal inflammation development.

Following this recommendation, INTRODUCTION has been simplified.

Results:1. It is recommended to add HE staining and immunohistochemistry staining to observe the inflammation, tissue damage, and repair status from 0 to 7 days, so that readers can understand cell phenotype changes corresponding to the periodontitis stage. The observation index can include inflammation and vascular related indicators.

As recommended, representative histological figures were included. We further performed new immunohistochemistry experiment of mouse gingival tissue (D0, D1, D3, D7) highlighting the infiltration of CD45+ immune cells. We found that inflammatory vascular formation in the H&E histology, which was highlighted. To characterize the tissue damage, the histological sections were stained by picrosirius red to highlight the change in collagen connective tissue of PDL and gingiva.

1. Figure 1A-1D can be placed in the supplementary figure.

Combining the new data above, Figure 1 was revised as suggested.

1. I suggest the authors to put the detection of the existence of AG fibroblasts before exploring its relationship with other types of cells.1. The layout of the picture should be closely related to the topic of the article. It is recommended to readjust the layout of the picture. Figure 1 should be the detection of AG cells and their proportion changes from 0 to 7 days. In other figures, the authors can separately describe the proportion changes of myeloid cells, T cells and ILCs, and explored the association between AG fibroblasts and these cell types.

As suggested, the presentation order of Figures and text was revised to bring the information about AG fibroblasts first. The chemokine-receptor analysis was moved below.

1. Please provide the complete form of "KT" in Line 162.

KT fibroblasts (fibroblasts keeping typical phenotype) was described in the text.

Methods:It is recommended to separately list the statistical methods section. The statistical method used in the article should be one-way ANOVA.

A separate statistical method section is created. As pointed out, we used one-way ANOVA with post-hoc Tukey test (when multiple groups were compared).

Discussion:I suggest the authors remove Figures 3-6 from the discussion section. For example, in Line 283, "(Figure 3 and 4)" should be removed.

Revised as suggested.

Reference:Some information for the references is missing. For example, "Lin P, et al. Application ofLigature-Induced Periodontitis in Mice to Explore the Molecular Mechanism of PeriodontalDisease. Int J Mol Sci 22, (2021)" should be "Lin P, et al. Application of Ligature-Induced Periodontitis in Mice to Explore the Molecular Mechanism of Periodontal Disease. Int J Mol Sci 22, 8900 (2021)". It is necessary to recheck all references.

The reference has been checked for the accuracy and the omission pointed out was corrected. Although we used EndNote program, we found some more inaccuracy in the references that were manually corrected. We appreciate your suggestion.

**Reviewer #2 (Recommendations For The Authors):**
1. Many host cells participate in immune responses, such as gingival epithelial cells. AG fibroblast is not the only cell involved in the immune response, and the weight of its role needs to be clarified. So the expression in the conclusion should be appropriate.

Following this critique, we revised INTRODUCTION, DISCUSSION and CONCLUSION, to highlight how AG fibroblasts function within a comprehensive immune response network.

1. This study cannot directly answer the issue of the relationship between periodontitis and systemic diseases.

We agree with this critique. We either deleted or de-emphasized the relationship between periodontitis and systemic diseases throughout the text.